

# Similarity and dissimilarity in model-results between single and multiple flow direction simulations based on a distributed ecohydrological model

Zhenwu Xu, and Guoping Tang

Department of Water Resources and Environment, School of Geography and Planning, Sun Yat-Sen University, Guangzhou, Guangdong, 510275, China

*Correspondence to*: Guoping Tang (tanggp3@mail.sysu.edu.cn)

**Abstract.** Distributed hydrological and ecosystem models route surface and subsurface flow in general two ways: single and multiple flow direction (SFD vs. MFD). However, few study has examined how model results differ between SFD and MFD
simulation. The objective of this study is to explore the similarity and dissimilarity of model results between the two simulations. To achieve this objective, we compared model results based on a distributed ecohydrological model that was ran respectively under four different routing algorithms. Our results indicate that SFD and MFD algorithms behave in the same way in simulating stream hydrograph. When averaged for the watershed, the values of modeled ecohydrological variables do not differ significantly between SFD and MFD simulations. Nevertheless, due to differences between SFD and MFD
algorithms, the modeled values of variables are spatially more autocorrelated under MFD algorithms than under SFD algorithms. In addition, there are significant differences in the modeled values of variables at individual cell level. For hydrological variables, differences are most significant in areas near channels and tend to decrease significantly as distances-from-channel increases. For ecological variables, differences are minimal in areas near channels and tend to increase significantly as distance-from- channel increases. The similarity and dissimilarity between SFD and MFD simulations are not
subject to dry or wet year. Overall, we found there are no differences in model results between SFD and MFD algorithms when model results are averaged for a study area but there are significant differences in model results when individual cells are compared.

## 1 Introduction

Surface and subsurface flow routing are important hydrological processes to the transportation of water, nutrients and
sediments in the land surface (Quinn et al., 1991; Orlandini & Moretti, 2009; Camporese et al., 2010). Process-based and distributed hydrological models such as DHSVM (Wigmosta et al., 1994) and RHESSys (Band, 1993; Tague & Band, 2004) as well as ecological models such as LPJ-GUESS (Smith et al., 2001; Tang et al., 2013) explicitly simulates flow routing to characterize the spatial patterns of surface and subsurface flow and their connection among simulated cells as well as to the drainage network (Jencso et al., 2009; Lane et al., 2009). It was found that the consideration of flow routing in distributed





models has important effects on accurate quantification of ecohydrological processes such as the evapotranspiration of hydrologic cycle (Tang et al., 2014) and the spatial patterns of saturated areas (Grabs et al., 2009). Thus, the selection of flow routing algorithms may be important for a model to accurately capture the upslope contributing areas, the sediment transport capacity, the topographic wetness and stream hydrograph in the land surface (Wilson et al., 2007; Camporese et al., 2010; Tang et al., 2015).

The existing flow routing algorithms in distributed models can be classified into two categories: single and multiple flow direction (SFD vs. MFD), both of which are built on digital elevation models (DEMs) and assume that flow basically follows the topographic relief. In practice, a 3-by-3 cell window is used to determine the direction of flow and the downslope cells that receive a portion of water. For SFD such as D8 (O'Callaghan & Mark, 1984) and D∞ (Tarboton, 1997), flow generated from a center cell will be routed to one or two downslope cells. For MFD such as MD8 (Quinn et al., 1991), DAEMON (Costa-Cabral & Burges, 1994), MD∞ (Seibert and McGlynn, 2007), TFN (Zhou et al., 2011), TFM (Pilesjö & Hasan, 2014), more flow directions are considered and more downslope cells will receive a portion of water. Due to differences in algorithms for routing flow along landscape, it becomes essential to understand how a model's performance may vary under different routing algorithms.

Previous studies have argued the limitation of SFD regarding the divergence in estimating flow paths (Quinn et al., 1991; Seibert and McGlynn, 2007) and the competence of MFD in producing better results in several topographic indices (e.g. specific catchment area, Wilson et al., 2007; Zhou et al., 2011; Pilesjö & Hasan, 2014). Erskine et al. (2006) found that the allowance of flow divergence on undulating terrains enable MFD to better estimate upslope contributing area at 5 m and 10 m grids. Kopecký and Čížková (2010) indicated that the MFD algorithms outperform SFD in capturing topographic wetness index (TWI) in vegetated area. Field observations at local scale indicate that SFD can result in an overestimate of runoff over low-flat regions as well as carbon uptake and release in the peatland (Tang et al., 2015). Nevertheless, previous assessments on flow routing algorithms are subject to the resolution of DEMs, which influence the spatial variability of DEM-derived slope, drainage area and TWI (Hasan et al., 2012). Furthermore, study based on different models or same model but differing model's parameterization is likely to bias the existing conclusions. Therefore, it is necessary to further assess how model results resemble or differ from each other among different routing algorithms when using a same model.

The objective of this study is to explore the similarity and dissimilarity in model results between SFD and MFD algorithms. To this end, a distributed ecohydrological model is applied to a typical mountain watershed in the western US. Simulations based on four different routing algorithms (two for SFD and two for MFD) and same model forcing data and model's parameterization were performed, respectively. Statistical methods – such as linear regression, spatial autocorrelation etc. – are used to analyze the similarity and dissimilarity in modeled values of ecohydrological variables among four routing algorithms at both watershed and individual cell level. The results of this study provide not only evidences on the effects of routing algorithms on model's performance but also insights into future efforts to improve model-based research in studying ecohydrological processes in the land surface.





## 2 Data and methods

### 2.1 The hydro-ecological simulation systems for simulating flow routing

The rasterized regional hydro-ecological simulation systems (hereafter renamed CHESS to stand for the coupled hydrology and ecology simulation systems, Tang et al., 2014, 2016) was selected for this study. CHESS was built on the Regional Hydro-Ecological Simulation Systems (Tague and Band, 2004) and designed for simulating integrated water, carbon, nutrient dynamics as well as vegetation growth in terrestrial ecosystems at watershed and regional scales. Because CHESS explicitly simulates lateral flow routing, it is suitable for testing how differences in flow routing algorithms may affect model results. Since specific algorithms for carbon, water, and nutrient dynamics in CHESS are mostly maintained as same as in Tague and Band (2004) and Tang et al. (2014, 2016), we herein briefly introduce the flow routing algorithms considered in this study.

Four different flow routing algorithms were considered in CHESS. Two routing algorithms are for SFD and the other two are for MFD. The first SFD algorithm is the traditional D8 algorithm (O'Callaghan and Mark, 1984), which was proposed and followed the prevalence of DEM and the necessity of simulating streamflow. The D8 algorithm only allows a downslope cell $i$ with maximum slope gradient to receive water (Fig. 1a). For D8, the portion of water $f_i$ to be distributed to the downslope cell $i$ is expressed as:

$$f_i = \begin{cases} 1 & (maximun\ tan\beta_i) \\ 0 & (others) \end{cases} \tag{1}$$

The first MFD algorithm considered in CHESS is the traditional MD8 algorithm (Quinn et al., 1991), in which all downslope cells will receive water and the portion of water received by a downslope cell $i$ is determined by its portion in the total slope gradient (Fig. 1b):

$$f_i = \frac{tan\beta_i}{\sum_{i=1}^{n} tan\beta_i} \tag{2}$$

where, $\beta_i$ is the slope gradient between the downslope cell i and the center cell M; n is the total number of downslope cells.

The second SFD algorithm considered in CHESS is proposed by Tarboton (1997) and named the D∞ algorithm, which uses the triangular facets to remove the limitation of only eight possible flow directions in the traditional D8. In order to describe infinite possible flow pathways, the D∞ algorithm assumed that a triangular facet with the steepest flow path receives all water. For triangular-facet-based algorithms, the first step is to calculate the steepest slope gradient in every triangular facet with one or two cells lower than the center cell. Then flow generated in the center cell $M$ is firstly divided to one (D∞) or several (MD∞) facets by a portion $T_k$, which will be further divided to one or two neighboring cells of the triangular facet determined by an angle (e.g. $\alpha_1$ in Fig. 1c).

Assume that $C$ and $D$ are two downslope cells. $h_1$ is the elevational difference between the center cell $M$ and the downslope cell C, and $h_2$ is the elevational difference between the downslope cell $C$ and the downslope cell $D$ (Fig. 1c):





$$\begin{cases} h_1 = e_M - e_C \\ h_2 = e_C - e_D \end{cases} \tag{3}$$

where, $e_M$, $e_C$ and $e_D$ are the elevation of cell $M$, $C$ and $D$, respectively. Assume $\alpha_1$ is the angle between the cardinal direction (e.g., the blue arrow MC in Fig. 1c) and the steepest slope direction (e.g., the red arrow in Fig. 1c). Then, the flow direction is determined by the following equation:

$$S_k \begin{cases} h_1/A, \alpha_1 = 0° & (h_2 \leqslant 0) \\ (h_1 + h_2)/(\sqrt{2}A), \alpha_1 = 45° & (h_2 \geqslant h_1) \\ \sqrt{h_1^2 + h_2^2}/A, \alpha_1 = \arctan(h_2/h_1) & (0 < h_2 < h_1) \end{cases} \tag{4}$$

where, $S_k$ is the steepest slope gradient in the triangular facet $k$; $A$ is the cell size.

Then, water is firstly assumed to flow to the facet $k$ and the portion $T_k$ is computed as:

$$T_k = \begin{cases} 1 & (maximun \ S_k) \\ 0 & (others) \end{cases} \tag{5}$$

Next, the water flowing to the triangular facet $k$ will be further redistributed as follows:

$$\begin{cases} f_{i,k} = T_k \alpha_2/45° \\ f_{j,k} = T_k \alpha_1/45° \end{cases} \tag{6}$$

where, $f_{i,k}$ and $f_{j,k}$ represent the portion of receiving water in cardinal direction $C$ and the diagonal direction $D$ (e.g., cell 6 and 7 in Fig. 1) of the triangular facet $k$. After computing all triangular facets, the portion of water $f_i$ to a receiving cell $i$ is eventually expressed as (D∞, Fig. 1c):

$$f_i = \sum_{i=1}^{n} f_{i,k} \tag{7}$$

where, $n$ is the total number of triangular facets with one or two downslope cells lower than middle cell $M$.

Our initial analysis indicated that the results of D∞ and MD∞ (Seibert and McGlynn, 2007) are almost identical in our pre-selected watershed largely because only 6.7% cells differ in flow routing between D∞ and MD∞ algorithm (Fig. S1). Given this, we slightly modified MD∞ and the new proposed algorithm is named RMD∞, which aims to combine the advantages of D∞ and MD8 and only differs from D∞ in computing $T_k$ (Eq. 5). Specifically, RMD∞ computes the steepest slope gradients in every triangular facet and divides water to every triangular facet by the portion of its steepest slope gradient in total steepest slope gradient as follows (Fig. 1d):

$$T_k = \frac{S_k}{\sum_{i=1}^{n} S_k} \tag{8}$$

where, $S_k$ is the steepest slope gradient in triangular facet k. After dividing water to every triangular facet, it repeats Eq. (6) and Eq. (7) to calculate the portion of water distributed to a receiving cell $i$.

## 2.2 Application of the distributed hydro-ecology simulation systems

We applied CHESS to the Cleve Creek watershed in the eastern Nevada of the US. This watershed is a typical semiarid to arid mountain watershed with slopes varying from 0.12° to 48.7°. With warm summers and cold winters, precipitation



mostly occurs in winter as snowfall. Because the watershed is located in the dry desert with spatially variable terrain and semiarid to arid climate, the ecohydrological processes are sensitive to water's availability. Therefore, it is an ideal laboratory to examine how the model results based on different routing algorithms may resemble and differ from each other. We run CHESS for the watershed respectively under each of the four routing algorithms.

## 2.3 Model forcing data and modeling design

The land cover data used to run CHESS are pre-specified for the watershed and consist of bare ground, conifer, grass and shrub (Fig. 2a). As in Tang et al. (2016), time series of daily minimum temperature, daily maximum temperature and daily total precipitation are also required to run CHESS. These data for the period 1961-2012 were derived from the Berry Creek SNOwpack TELemetry (SNOTEL) weather station, which is located 10 km away from the study area. Taking into account the elevational gradients between the SNOTEL station and the watershed, a local lapse rates of -0.0068 $℃m^{-1}$ for daily maximum temperature, -0.0045 $℃m^{-1}$ for daily minimum temperature and 0.001 $mm\ m^{-1}$ for daily precipitation were used to adjust effects of topography on temperature and precipitation in the study area.

To minimize the effects of other factors on model results under the four routing algorithms, we carefully designed model simulations: first, the same model forcing data were used to run CHESS under each of four routing algorithms; Second, we carefully calibrated and evaluated model simulations so that the modeled streamflow under each of four routing algorithms approximates the observed value as possible as it can; Third, we kept almost all model parameterizations identical under the four routing algorithms while only slightly adjusted the values of soil hydraulic conductivity decay parameter in each of four simulations. By doing so, differences in model results among the four routing algorithms can thus be attributable to differences in flow routing algorithms themselves. We calibrated model simulations for the period 1991-2000 and evaluated them for the period 2001-2012 using observed streamflow and derived baseflow from one US Geological Survey station (Fig. 2).

## 2.4 Statistical analysis

Given CHESS is a process-based distributed model that simulates ecohydrological processes at individual cell level, it's crucial to compare the pattern of modeled ecohydrological processes at both watershed and individual cell level. Population analysis can take all cell-based values as an integrated sample and calculates some important indices – such as mean and standard deviation – to acquire the population distribution. Likewise, spatial analysis is useful for examining the spatial distribution and autocorrelation of simulated ecohydrological variables. These analytical methods are used in this study.

In specific, the Global Moran's index (Moran, 1948) is used to quantify the similarity and dissimilarity in the spatial patterns of modeled variables among the four routing algorithms. Theoretically, this index aims to help understand the degree of similarities of nearby locations in space. It measures the spatial autocorrelation of a variable and expresses the homogeneity of a variable in its neighborhood extent (Dray et al., 2006). Moran's I is widely used in spatial analysis (Garcia-Baquero et al., 2016; Smouse and Peakall, 1999) and expressed as:



$$I = \frac{n \sum\limits_{i=1}^{n} \sum\limits_{j=1}^{n} w_{i,j} z_i z_j}{\sum\limits_{i=1}^{n} \sum\limits_{j=1}^{n} w_{i,j} \sum\limits_{i=1}^{n} z_i^2} \tag{9}$$

where, $w_{i,j}$ is the weight between the observation $i$ and j; $n$ is the total number of grids; and $z_i$ is the difference between the value of i and the mean value of a variable.

Besides, we followed Tesfa et al. (2011) to have created a distance-to-stream map (Fig. 2b) to measure the shortest horizontal distance from a cell to channels. The map is used to measure how differences in model results between a pair of compared algorithms change over space in terms of cell's distances to channels. To measure the differences between SFD and MFD algorithms, the relative deviation $D_R$ between modeled values of a variable in a cell is calculate as:

$$D_R = \frac{|I_{SFD} - I_{MFD}|}{(I_{SFD} + I_{MFD})/2} \tag{10}$$

where, $I$ is for an ecohydrological variable. To obtain a single value of $D_R$ for all cells within a certain distance from stream channel, we averaged all cell-based $D_R$ values within the given distance as follows:

$$D_{AR} = \sum\limits_{i=1}^{n} D_R / n \tag{11}$$

where, d is for a certain distance to stream; n is the total number of patches at this distance. The averaged relative deviation ($D_{AR}$) is used to help us understand how model results between two different routing algorithms will vary as distance from channel increases.

## 3 Results

### 3.1 Equivalency in model's performance in simulating stream hydrograph

Our results indicate that all simulations under the four routing algorithms can be calibrated to accurately simulate observed streamflow and derived baseflow from the gauge station in an extremely similar manner (Table 1). For example, differences in calculated Nash-Sutcliffe coefficients (NSs) under the four algorithms are less than 0.1% for both streamflow and baseflow in the calibration period. For the evaluation period, differences in NSs are also less than 0.1% among the four algorithms. In addition, the NSs for streamflow are 0.75 for the calibration period and 0.81 for the evaluation period under all routing algorithms, indicating that all simulations under differing routing algorithms are satisfactory. For instances, the simulated annual mean daily streamflow and baseflow not only agree well with observed and derived values, respectively, but also are highly consistent with each other (difference < 0.1%) among the four algorithms (Table 1). The root-mean-square errors (RMSE) between simulated and observed streamflow as well as between simulated and derived baseflow are less than 0.1%. These similarities suggest that the four algorithms performed in the same way in accurately capturing stream hydrograph of the watershed (e.g., Fig. S2).

When it comes to model's performance on an annual basis, although there is a slight variation in the calculated NSs among the four routing algorithms, differences in NSs are not statistically significant (< 5%). For example, the simulated





annual mean streamflow is extremely similar to each other for all years among the four routing algorithms (Table A1). Besides, when it comes to a specific dry or wet year, there exists no significant difference in the magnitudes of NSs among the four algorithms. For example, the NSs are almost identical for the wet year of 2005 (0.84 for D8, 0.85 for MD8, 0.85 for D∞, and 0.86 for RMD∞) and the dry year of 1992 (0.93 for D8, 0.93 for MD8, 0.92 for D∞, and 0.93 for RMD∞) (Table S1). These equivalents indicate that the four routing algorithms performed equivalently in capturing stream hydrograph in the study area, and the model's performances under the four routing algorithms are not sensitive to the amount of precipitation. Besides, evaluation on ecological variables (e.g., leaf area index and net primary productivity) is conducted as in Tang et al. (2016) and presented in the supplementary materials, indicating that CHESS captured well the spatial and temporal patterns of LAI and NPP in the study area.

**3.2 Equivalency in model's behavior at the watershed scale**

Table 2 shows the comparisons of simulated soil saturation deficit (SSD) and leaf area index (LAI) at watershed scale among the four routing simulations, respectively. When averaged for the watershed, there are no significant differences in the values of modeled SSD and LAI, respectively, among the four algorithms. For example, the range and mean of modeled SSD are almost identical (difference < 0.1%) among the four simulations (Table 2). Similarly, differences in minimum, maximum and mean LAI also are less than 0.02 unit among the four simulations. The values of standard deviation for both SSD (difference < 5%) and LAI approximate (difference < 0.015 $m^2 m^{-2}$) each other respectively among the four simulations (Table 2), indicating that there are no significant differences in the dispersion of modeled annual mean SSD and LAI values when averaged for the watershed. These similarities indicate that the selection of the four routing algorithms has the least impact on the ranges and dispersions of modeled values of ecohydrological variables when averaged for the watershed.

**3.3 Dissimilarity in spatial autocorrelation of modeled ecohydrological variables**

Table 3 shows the spatial autocorrelation (measured by the Global Moran's index) of modeled SSD and LAL among the four simulations for the dry year 1992, the wet year 2005 and the whole study period, respectively. The statistics indicate that there exist apparent similarity and dissimilarity in the spatial autocorrelation of modeled ecohydrological variables. First, the similarities in spatial autocorrelation of modeled variables are higher between the two SFD or MFD algorithms when compared to those between SFD and MFD algorithm. For example, difference in Moran's I for SSD between D8 (0.425) and D∞ (0.436) is less than 3%, suggesting that the autocorrelation of modeled SSD is extremely similar between the two SFD algorithms. This situation also applies to the two MFD algorithms, under which the Moran's I agrees well between MD8 (0.515) and RMD∞ (0.494) (Table 3), suggesting that there is no significant difference (< 4%) in the spatial autocorrelation of modeled SSD between the two MFD algorithms. This similarity applies to not only modeled LAI but also both the dry and wet year (Table 3).



Second, when comparisons were made between SFD and MFD algorithms, the values of modeled variables under MFD algorithms are spatially more autocorrelated than those under SFD algorithms. For example, the Moran's I for SSD is 0.515 under MD8 for the calibration period, which is 21% higher than that (0.425) under D8 (Table 3). Similarly, the Moran's I for SSD under RMD∞ (0.494) is 13% higher than that (0.436) under D∞ for the calibration period (Table 3). This distinct contrast (higher for MFD algorithms and lower for SFD algorithms) also applies to not only modeled LAI but also both the dry and wet year (Table 3). Nevertheless, the dissimilarity in the spatial autocorrelation of modeled LAI becomes weaker (difference ranges from 8% to 9%) when compared to that of SSD (difference ranges from 13% to 21%) between SFD and MFD algorithms.

### 3.4 Differences in the patterns and magnitudes of simulated hydrological variables at cell level

Because the modeled values of ecohydrological variables among the four routing algorithms are almost equivalent when averaged for the watershed and because differences in modeled values of variables are only statistically significant between SFD and MFD algorithms, our further comparisons of the patterns and magnitudes of modeled values targeted only results between D8 and MD8 algorithm. Figure 3 shows the spatial pattern of modeled SSD under D8 and MD8 algorithm as well as differences in SSD between the two algorithms in a dry (1992, Fig. 3a,b,c) and wet year (2005, Fig. 3d,e,f). Regardless of the magnitudes of simulated SSD at cell level, the Kappa statistics (0.64 for SSD in 1992 and 0.63 for SSD in 1995; Note: we converted numerical data to categorical data based on pre-specified ranges of values for calculating the Kappa statistics) indicate a very good agreement in the spatial pattern of modeled SSD between the two algorithms. For example, SSD is modeled to be low in areas near channels while high in the ridges or hills of the watershed (Fig. 3a,b). In the southeastern corner of the watershed, SSD is high under both D8 and MD8 in the 1992-dry year (Fig. 3a,b), reflecting the reality where soil tends to be drier as temperature increases down to low elevation areas. This kind of similarity in the spatial patterns of modeled SSD also applies to the 2005-wet year (Fig. 3d vs 3e). For example, the spatial pattern of modeled SSD in the 2005-wet year is extremely similar between D8 and MD8 algorithm: lower in areas near channels and higher in the ridges or hills of watershed (Fig. 3d, 3e). Differences in modeled SSD between the dry and wet year lie in that the areas with lower SSD near channels in the wet year are broader than those in the dry year (e.g., Fig. 3b vs. 3e).

Nevertheless, when it comes to the magnitudes of simulated SSD at cell level, there exist significant differences in modeled SSD between D8 and MD8 algorithm in both dry and wet year. For example, differences in the magnitudes of modeled SSD are mostly significant (difference > 5% or < -5%) in areas near channels in the 1992-dry year (Fig. 3c). For areas far away from channels, such differences are not significant (Fig. 3c). This phenomenon also applies to modeled SSD between the two algorithms in the 2005-wet year (Fig. 3f). Further analyses indicate that differences in modeled SSD between D8 and MD8 algorithm decreases significantly as the distances of cells to channels increases (Fig. 4a,b). Furthermore, differences in SSD between D8 and MD8 algorithm in the 1992-dry year are strongly significantly correlated with those in the 2005-wet year (Fig. 4c), indicating that the mode of differences in SSD between D8 and MD8 algorithms are not subject to the amount of precipitation.



### 3.5 Difference in the patterns and magnitudes of simulated ecological variables at cell level

Figure 5 shows the spatial pattern of modeled LAI under D8 and MD8 algorithm as well as differences in LAI values between the two algorithms for a dry (1992) and wet (2005) year, respectively. Like SSD, regardless of the magnitudes of LAI at cell level, the Kappa statistics (0.70 for both the dry and wet year) indicate a very good agreement in the spatial pattern of modeled LAI between D8 and MD8 in both the dry and wet year (Fig. 5a vs. 5b; 5d vs. 5e). For example, LAI for the dominant vegetation shrub is modeled to be higher in areas near channels and lower in the ridges or hills of the watershed for the 1992-dry year (e.g., Fig. 5a, b; note: the highest LAI occurs in conifers, which somewhat mask this kind of patterns). This phenomenon also applies to the wet year, in which the modeled LAI is still higher at valleys than at ridges of the watershed (Fig. 5d, e). Overall, the spatial pattern of modeled LAI is extremely similar between D8 and MD8 algorithm and this similarity is not subject to the amount of precipitation in a year.

However, when it comes to differences in modeled LAI values between D8 and MD8 algorithm, there are similarity and dissimilarity between the two algorithms. In the 1992-dry year, for instance, the modeled LAI values in areas near channels are extremely similar (-5% < difference < 5%) between D8 and MD8 algorithm (Fig. 5c). In contrast, there are significant differences (> 5% or < -5%) in LAI values in areas away from channels between the two algorithms (Fig. 5c). Further analysis indicates that differences in modeled LAI values first increase, then become stable and again increase as the distance from channel increases (Fig. 4d,e). Such similarity and dissimilarity in modeled LAI values also apply to the wet year, indicating that the difference in the patterns of modeled LAI values is not sensitive to the amount of precipitation. The scatter plot (Fig. 4f) further suggests that differences in modeled LAI values in the 1992-dry year are significantly correlated with those in the 2005-wet year between D8 and MD8 algorithm.

## 4 Discussion

### 4.1 Equivalence in model's performance in simulating ecohydrological processes at watershed scale

The equivalence in model's performance under the four routing algorithms in simulating stream hydrograph attributes partially to model's calibration, by which we keep streamflow in the outlet as close as the observed value under all routing algorithms. In addition, for stream-type cells, differences in accumulated area among the four routing algorithms tend to become smaller and eventually zero as water moves from the head to the outlet of watershed. This explains why the modeled streamflow turns almost identical among the four algorithms (Fig. 6). Compared to other studies without strict control of outlet streamflow to a same level (Tang et al., 2015), our study suggests the selection of routing algorithms has the least effects on the model's performance in simulating stream hydrograph at the outlet of the watershed because the accumulated areas for the outlet is a constant under all routing algorithms.

In terms of the equivalence in modeled values of ecohydrological variables when averaged for the watershed, it can be explainable from the perspective of mass balance. Because model forcing data (especially precipitation) are identical among





the four algorithms and because we have modeled streamflow close to the observed value under each of four algorithms, it is not surprising that the modeled SSD approximates each other among the four routing algorithms when averaged for the watershed. For example, a previous study (Wolock and McCabe, 1995) indicated that any difference in the efficiency and simulated flow paths of TOPMODEL between the SFD and MFD algorithms essentially disappeared when the model was
well-calibrated.

For LAI, because vegetation growth depends mainly on soil water content and temperature while temperature is kept identical among the four routing simulations, the greater the SSD is, the smaller the LAI tends to be, and vice versa. Thus, differences in modeled LAI between SFD and MFD algorithm can offset each other at individual cell level. Thus, when averaged for the watershed, the simulated values of LAI are almost identical between SFD and MFD algorithm. Overall, these
equivalences suggest that the model's performance in simulating stream hydrograph and the values of ecohydrological variables at watershed scale are not subject to the type of routing algorithms.

### 4.2 Differences in the spatial autocorrelation of simulated values

Our calculated Moran's I that ranges from 0.41 for SSD to 0.52 for LAI at 100 m resolution agrees well with Cai and Wang (2006). The higher spatial autocorrelation of modeled values under MFD simulations than under SFD simulations results
from differences in the dispersion of water, which is greater under MFD simulations than under SFD simulations. In fact, there exist distinct differences in the distribution of the number of cells where the generated flow from a center cell is routed to from 1 to 8 downslope neighbors. Under the D8 and D∞ algorithms, flow generated from a center cell is routed to only one or at most two downslope cells (Fig. 7). In contrast, flow generated from a center cell is mostly routed to three to five downslope cells under the MD8 and RMD∞ algorithms (Fig. 7). This implies that the modeled values of ecohydrological variables at a
given cell is more directly or indirectly affected by its neighbors under MFD algorithm than under SFD algorithm. This explains why the modeled SSD and LAI are spatially more autocorrelated under the MD8 and RMD∞ algorithms than those under the D8 and D∞ algorithms (Table 3).

### 4.3 Differences in simulated hydrological variable at individual cell level

The decreasing tendency of differences in SSD between D8 and MD8 algorithm as the distance of cells to stream increases
results largely from differences in accumulated area of flow at cell level. For the watershed outlet, the accumulated area of flow is a constant (8230 cells) among the four routing algorithms because all flow generated from the watershed are converged to the outlet. However, for other non-stream-type cells, the accumulated area of flow varies among cells. Further analyses indicate that differences in accumulated area of flow between D8 and MD8 algorithm decreases significantly as the distances of cells to channels increases (Fig. 8a). Thus, it is not surprising that differences in SSD between D8 and MD8 algorithm
showed the same tendency. This tendency of differences in SSD also applies to D∞ and RMD∞ algorithm (Fig. 8b) and is consistent with previous study. For example, Seibert and McGlynn (2007) indicated that relative differences in accumulated



area of flow between D8 and MD8 occur more apparently in lower region near the channels. In fact, there are about 64.5%/34.2% (for MD8) and 71.5%/22.8% (for RMD∞) of cells receive more/less water when compared to their counterparts under two SFD algorithms. The underlying reasons for such differences result from differences in the direction and dispersion of cells receiving flow between SFD and MFD simulations (Fig. S4).

## 4.4 Differences in simulated ecological variable at individual cell level

Spatial variations in soil moisture condition can have significant effects on canopy photosynthesis and forest productivity particularly under drying conditions (Band et al., 1993; Hwang et al., 2012). Differences in simulated LAI between SFD and MFD simulations are smaller in areas near channels are largely because soil water content in these areas are higher than areas far away from channels. In fact, the modeled SSD increases significantly as the distance from channels increases (Fig. 9a, b). As a result, the limitation of soil water condition on vegetation growth is minimal in areas near channels under both SFD and MFD simulations. In contrast, for areas near the ridges or hills of the watershed, because SSD is higher, the limitation of soil water condition on vegetation could be higher. Thus, any tiny changes in soil water condition may imply a big variation in vegetation growth such as leaf expansion. This explains why differences in simulated LAI between SFD and MFD simulation are greater in areas near the ridges or hills of the watershed.

## 5 Conclusions

In this study, we used a distributed ecohydrological model to have examined how the model results based on SFD algorithms may resemble or differ from those based on MFD algorithms. Our results indicate that:

1) A model based on either SFD or MFD algorithm can perform equivalently in simulating stream hydrograph of a watershed. When averaged for a study area, the modeled values of ecohydrological variables do not differ significantly between SFD and MFD simulations. This equivalency in model's performance between SFD and MFD algorithm applies to both dry and wet year.

2) The modeled values of ecohydrological variables are spatially more autocorrelated under MFD simulation than under SFD simulation. This discrepancy results from the dispersion of flow that is greater under MFD algorithm than under SFD algorithm, and also applies to both dry and wet year.

3) At individual cell level, there exist significant differences between SFD and MFD simulation. For hydrological variables, differences in modeled values tend to decrease significantly as the distance of cells from channel increases. This phenomenon is reversed for ecological variables. Again, the dissimilarity in modeled values between SFD and MFD algorithm are applicable to both dry and wet year.

Overall, our results indicate that the selection of routing algorithms has no effects on model results when they are averaged for the study area and differences in model results only occur at individual cell level. Therefore, spatial observations are essential for testing which routing algorithm better captures the reality of local ecohydrological processes, especially in



topographically variable terrain.

**Acknowledgement**

This study was supported by the Chinese National Natural Science Foundation (#41671192) and the Guangzhou Municipal
Scientific Program (#42050441). Daily weather records used in this study were downloaded from the Western Regional
Climate Center at the Desert Research Institute in Reno, Nevada (http://wrcc.dri.edu/).

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





## Figures

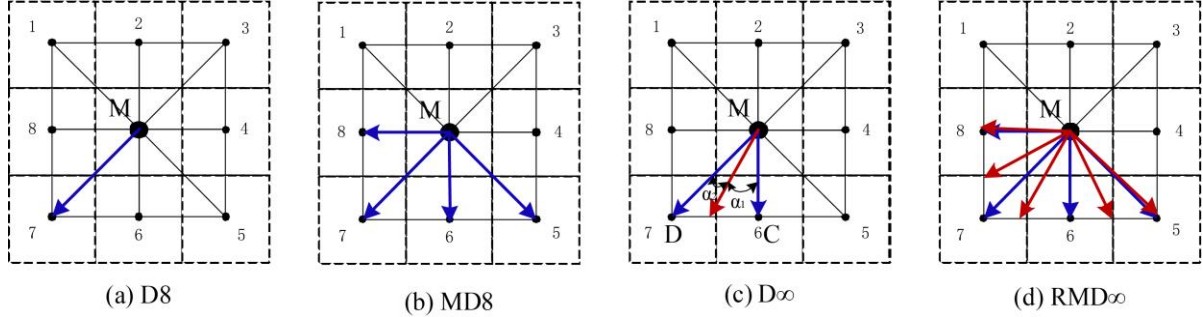

**Figure 1.** A conceptual map of D8, MD8, D∞ and RMD∞ algorithms considered in this study. The blue arrows represent the flow patch/direction in a 3-by-3 cell window. Cell 5, 6, 7 and 8 are assumed to be lower in elevation than that of the center cell M, and cell 7 has the steepest slope gradient. The red arrow refers to the steepest slope gradient in a triangular facet (e.g., M, 6, 7 in D∞), that consists of MC (the cardinal direction) and MD (the diagonal direction) in triangular-facet-based algorithms (D∞ and RMD∞).α

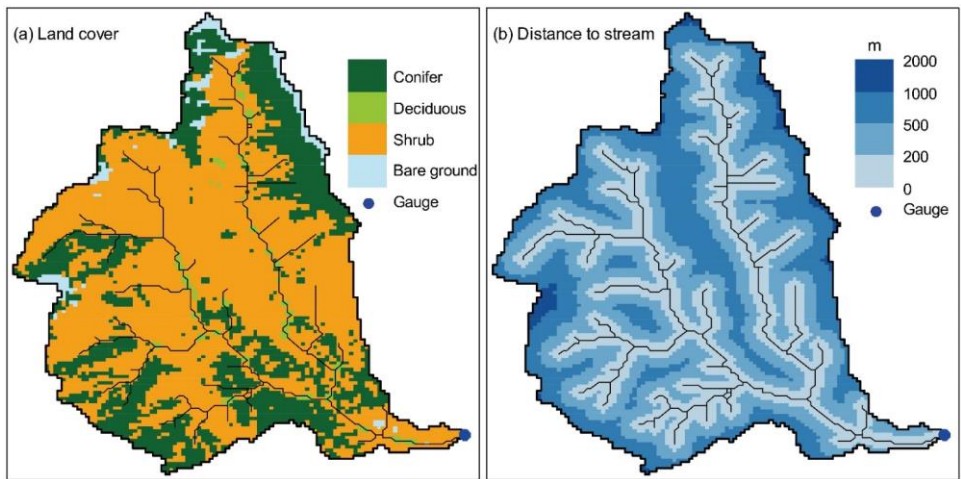

**Figure 2.** (a) Distribution of land cover in the Cleve creek watershed and (b) a map of the shortest horizontal distances to channels calculated for all cells.





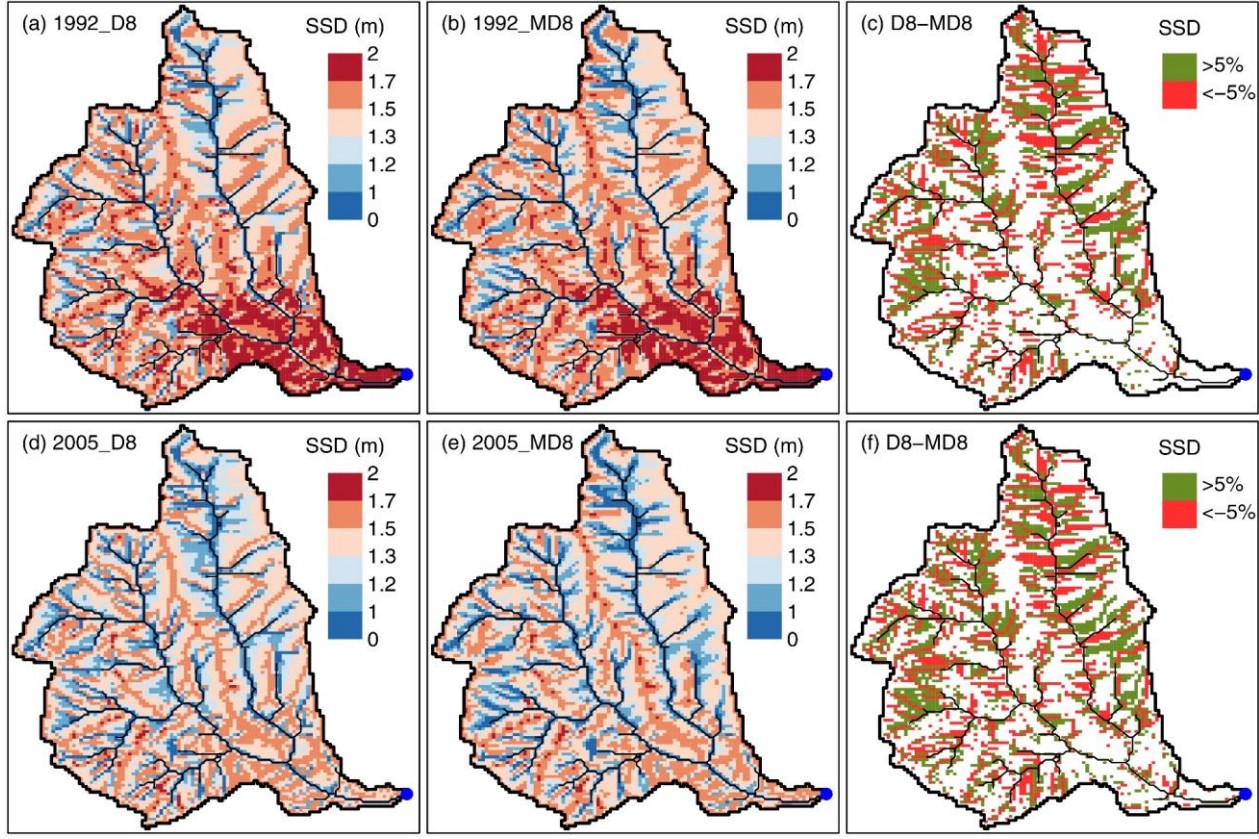

**Figure 3.** The spatial patterns of simulated soil saturation deficit (SSD) under (a, d) D8 and (b, e) MD8 algorithm; (c, f) the spatial pattern

of differences in SSD between D8 and MD8 algorithm.



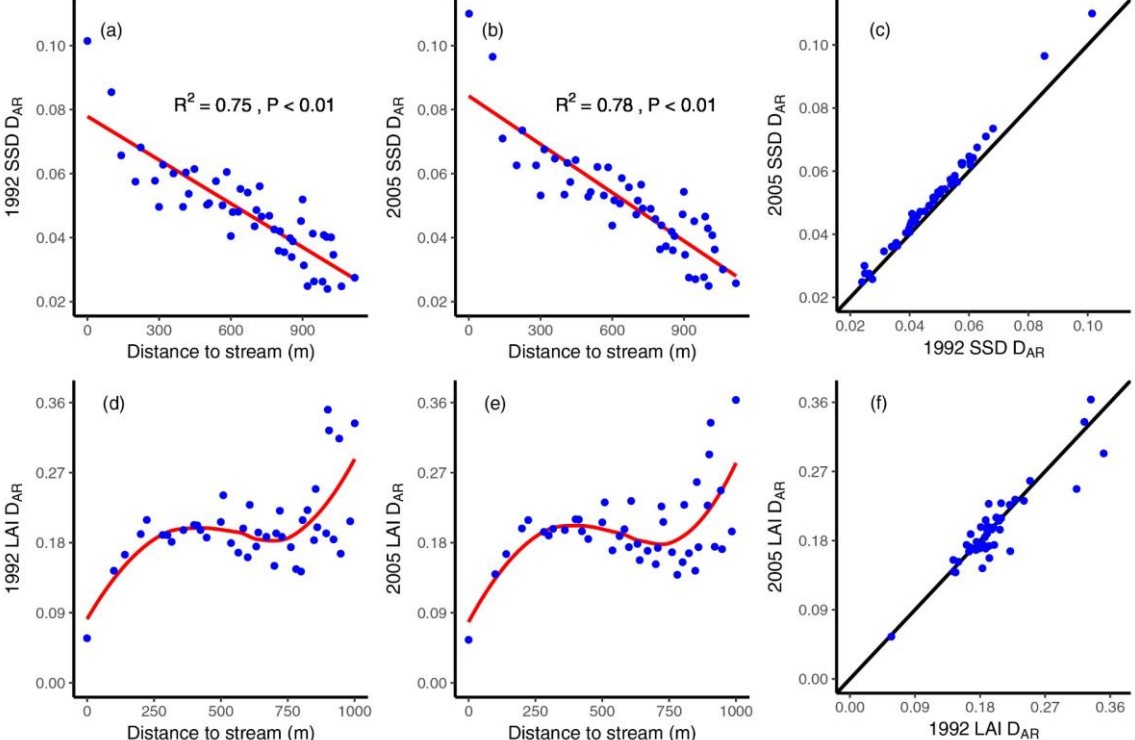

**Figure 4.** Changes in the average relative deviation ($D_{AR}$) of soil water saturation deficit (SSD) and leaf area index (LAI) as distances from channel increase under D8 and MD8 algorithm for the 1992-dry and 2005-wet year.





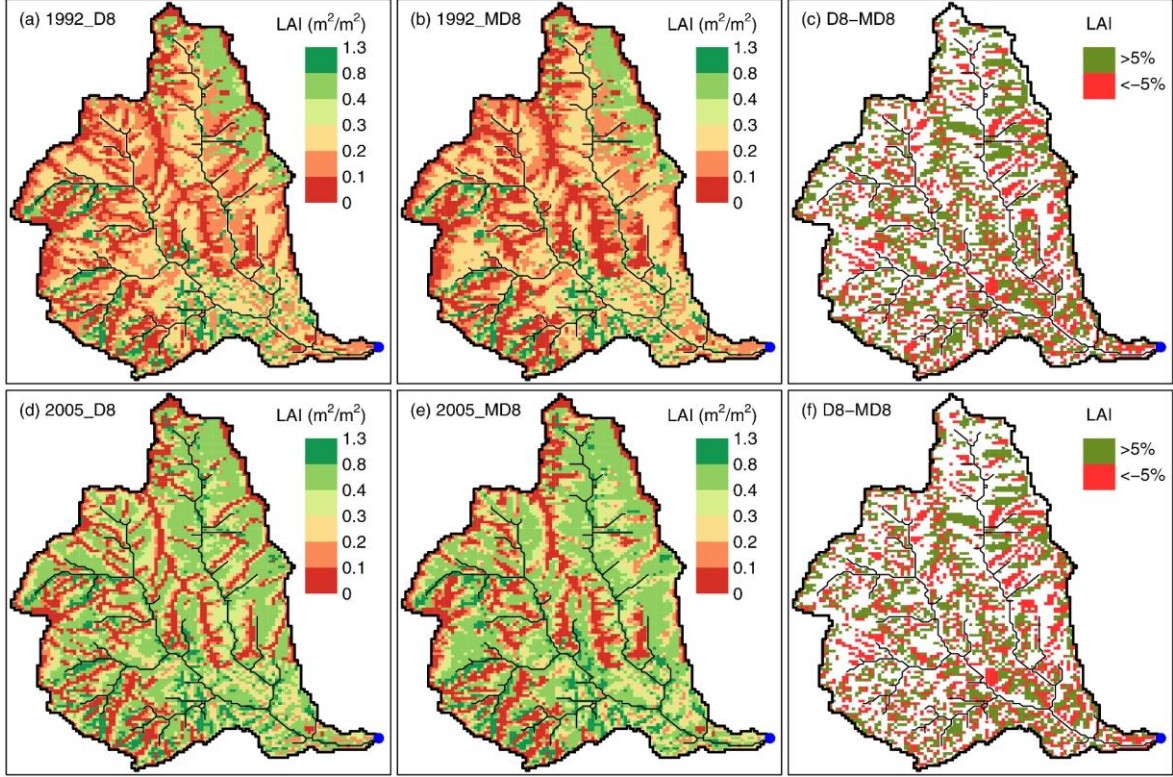

**Figure 5.** The spatial patterns of simulated leaf area index (LAI) under (a, d) D8 and (b, e) MD8 algorithm; (c, f) the spatial pattern of differences in LAI between D8 and MD8 algorithm.





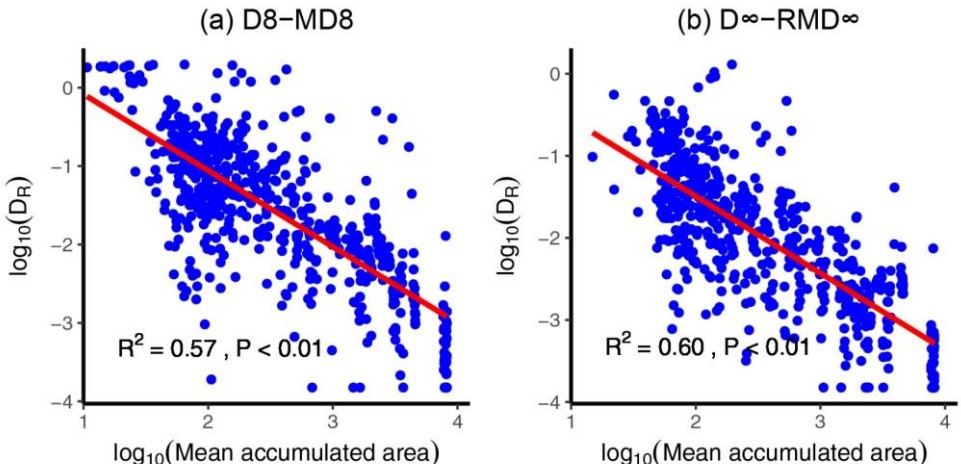

**Figure 6.** For stream-type cells, relative deviations ($D_R$) in accumulated area of flow between SFD and MFD simulations decreases apparently as flow moves from the head to the outlet of the watershed.

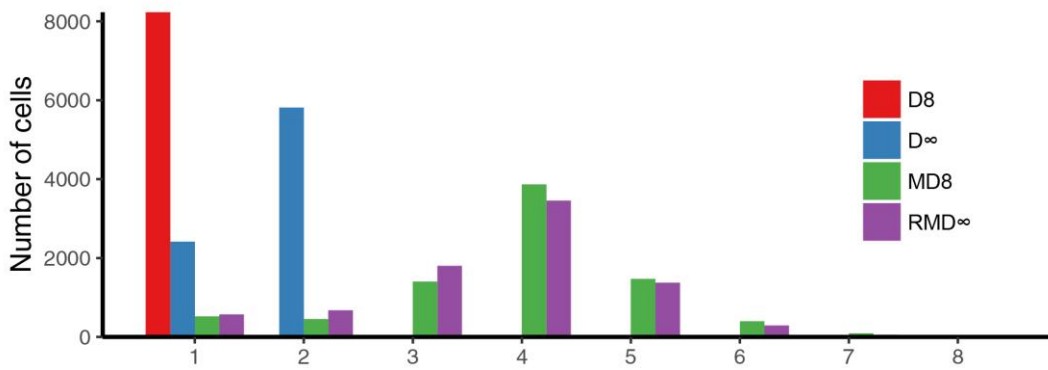

**Figure 7.** The total number of cells where flow generated from a center cell is routed to from 1 to 8 downslope neighbors. The digits in x-axis refers to the number of downslope neighbors receiving water from the center cell.



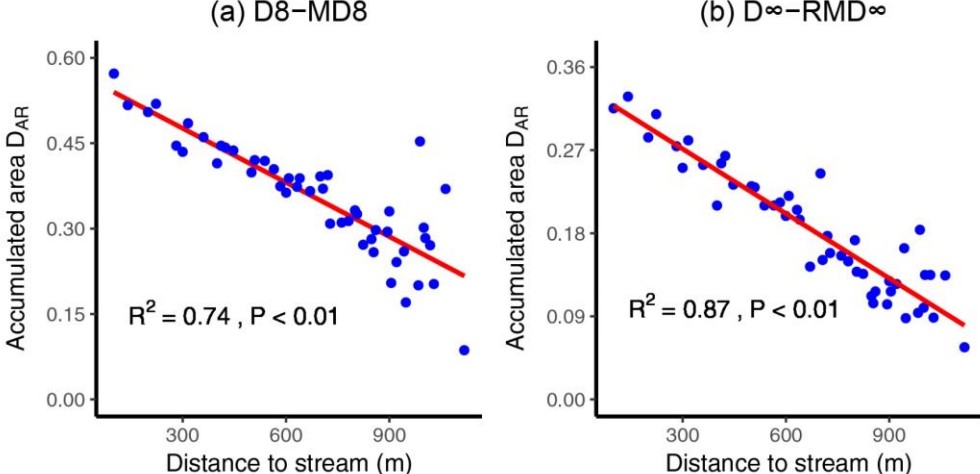

**Figure 8.** Changes in the average relative deviation ($D_{AR}$) of accumulated area of flow between SFD and MFD simulations (e.g., panel a: D8 vs. MD8; panel b: D∞ vs. RMD∞) increases significantly as the distance of cells to channels increases.

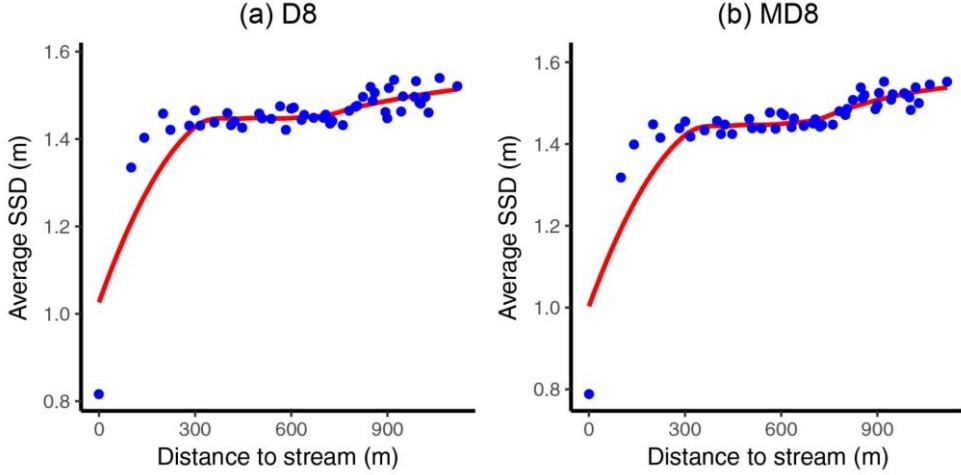

**Figure 9.** Average soil saturation deficit (SSD) under D8 and MD8 algorithm decreases significantly as the distance of cells to channels increases.



**Tables**

**Table 1.** Calibration and evaluation of model simulations under the four routing algorithms

| Algorithms | | | D8 | MD8 | D∞ | RMD∞ | σ |
|---|---|---|---|---|---|---|---|
| Outlet flow (mm/day) | 91-00 | SF | 0.596 | 0.596 | 0.596 | 0.596 | 0.000 |
| | | BF | 0.585 | 0.588 | 0.587 | 0.588 | 0.001 |
| | 01-12 | SF | 0.522 | 0.522 | 0.524 | 0.522 | 0.001 |
| | | BF | 0.508 | 0.522 | 0.511 | 0.521 | 0.006 |
| RMSE (mm/day) | 91-00 | SF | 0.270 | 0.271 | 0.271 | 0.271 | 0.000 |
| | | BF | 0.230 | 0.228 | 0.233 | 0.230 | 0.002 |
| | 01-12 | SF | 0.326 | 0.329 | 0.327 | 0.326 | 0.001 |
| | | BF | 0.284 | 0.282 | 0.282 | 0.281 | 0.001 |
| NS | 91-00 | SF | 0.749 | 0.748 | 0.748 | 0.748 | 0.001 |
| | | BF | 0.746 | 0.751 | 0.740 | 0.746 | 0.004 |
| | 01-12 | SF | 0.812 | 0.808 | 0.811 | 0.811 | 0.002 |
| | | BF | 0.823 | 0.826 | 0.826 | 0.827 | 0.001 |

*σ is standard deviation; RMSE is short for root mean squared error; NS is short for Nash-Sutcliffe coefficient; "91-00" and "01-12" represent the calibration period 1991-2000 and the evaluation period 2001-2012, respectively; SF is streamflow; BF is baseflow.

**Table 2.** Comparisons of modeled soil saturation deficit and leaf area index among the four routing algorithms averaged for the watershed

| | Soil saturation deficit (m) | | | | Leaf area index $(m^2 m^{-2})$ | | | |
|---|---|---|---|---|---|---|---|---|
| | D8 | MD8 | D∞ | RMD∞ | D8 | MD8 | D∞ | RMD∞ |
| Min | 0.058 | 0.036 | 0.054 | 0.038 | 0.000 | 0.000 | 0.000 | 0.000 |
| Max | 1.856 | 1.905 | 1.840 | 1.879 | 1.252 | 1.253 | 1.254 | 1.256 |
| Mean | 1.387 | 1.383 | 1.389 | 1.384 | 0.294 | 0.308 | 0.286 | 0.303 |
| σ | 0.290 | 0.283 | 0.290 | 0.285 | 0.249 | 0.252 | 0.239 | 0.246 |

*Statistics are calculated based on annual mean daily values averaged for the watershed for the study period.



**Table 3.** Comparisons of the spatial autocorrelation (measured by Moran's I) of modeled values among the four routing algorithms

| Algorithms | Soil saturation deficit | | | Leaf area index | | |
|:---:|:---:|:---:|:---:|:---:|:---:|:---:|
| | 1992 | 2005 | 1991-2012 | 1992 | 2005 | 1991-2012 |
| D8 | 0.418 | 0.414 | 0.425 | 0.417 | 0.436 | 0.424 |
| MD8 | 0.507 | 0.500 | 0.515 | 0.456 | 0.474 | 0.463 |
| D∞ | 0.433 | 0.425 | 0.436 | 0.414 | 0.431 | 0.419 |
| RMD∞ | 0.487 | 0.479 | 0.494 | 0.445 | 0.462 | 0.451 |
| p | $< 0.01$ | | | | | |