# Peer review of "Similarity and dissimilarity in model-results between single and multiple flow direction simulations based on a distributed ecohydrological model"

_Hydrology and Earth System Sciences, 2018_

## Short Comment (SC1) · 22 Feb 2018

This manuscript compared the effects of four flow direction algorithms (D8, D-inf, MD8, and MD-inf) applying to a distributed ecohydrological model, CHESS. I would suggest authors' possible consideration on one more multiple flow direction (MFD) algorithm, i.e. MFD-md proposed by Qin et al. (1997), which is adaptive to local terrain condition. MFD-md can achieve more accurate results than D8, D-inf, and MD8, based on evaluations with different artificial DEMs and their theoretical specific catchment area distributions (Qin et al., 2013). And one of recent evaluation researches on effects of

different flow direction algorithms on calculating topographic wetness index (TWI) and explaining soil moisture in real catchments (Radula et al., 2018) also showed that the strongest relationship between TWI and soil moisture was obtained by MFD-md, compared to several existing algorithms (such as D8, D-inf, MD-inf, MD8, Rho8, DEMON, KRA, BR, MFM, as well as Ellenberg's indicator values).

Qin C, et al. An adaptive approach to selecting a flow-partition exponent for a multiple-flow-direction algorithm. International Journal of Geographical Information Science, 2007, 21(4): 443-458.

Qin C-Z, et al. Artificial surfaces simulating complex terrain types for evaluating grid-based flow direction algorithms. International Journal of Geographical Information Science, 2013, 27(6): 1055-1072.

Raduła, et al. Topographic wetness index explains soil moisture better than bioindication with Ellenberg's indicator values. Ecological Indicators, 2018, 85: 172-179.

―――――――――――――――――――――

---

## Author Comment (AC1) · 17 Mar 2018

Dear Dr. Qin,

We greatly appreciate your valuable comments on our manuscript (discussion version: hess-2018-47). We have run model simulation based on the "MFD-md" algorithm and compared simulation results with those based on other flow routing algorithms. Our results suggest that: 1) All flow routing algorithms (including MFD-md) performed in an extremely similar way in simulating stream flow at the watershed outlet. When aver-
aged for the watershed, the modeled values of eco-hydrological variables are also similar among all routing algorithms, including MFD-md. We added additional results based on MFD-md in the supplements (e.g., Table. 1, 2 and 3). 2) In addition, differences in simulated values of ecohydrological variables between "MFD-md" and SFD (e.g., D8) algorithm also show same pattern/tendency as we discovered in the manuscript. For hydrological variables, differences at cell level tend to increase as the distances of cells from channel increase. For the modeled values of ecological variables, this tendency is reversed. We also added relevant results in the supplements (e.g., Fig. 1, Fig. 2 and Fig. 3) Overall, our results still indicate that difference in simulated values of eco-hydrological variables between MFD and SFD algorithms mainly occur at individual cell level. This is applicable when comparing "MFD-md"-based results with other SFD-based results. However, due to lack of field observed data, it is hard to conclude if a routing algorithm is superior to another, which is somewhat beyond the scope of our existing study but could be part of our future relevant research. Again, we highly appreciate your valuable comments on our manuscript. We will definitely cite your great work in the revised manuscript (if there is a chance). Best regards,

Zhenwu and Guoping

Please also note the supplement to this comment:
https://www.hydrol-earth-syst-sci-discuss.net/hess-2018-47/hess-2018-47-AC1-supplement.pdf

---

## Referee Comment (RC1) · Anonymous Referee #1 · 29 Mar 2018

This ms compared results from simulations of RHESS (authors renamed it to CHESS) model among four different flow routing algorithms. The authors showed that results from different algorithms differ at the individual cells but that the results for the whole watershed are the same. In my view, this is a trivial result, as – by definition – all cells of the watershed are drained through the same point, the results at the watershed scale should be the same regardless flow routing algorithm used. Many previous studies also showed that the results largely differ among flow routing algorithms at the individual cell. Therefore, this is also not novel finding. For me, the most novel finding presented

in the ms is proposed modification to MDinf algorithm of Seibert and McGlynn (2007). However, to be useful, the authors should show its utility by comparing it both to the field measured data and simulated data with known properties.

I do not agree with the authors that Dinf algorithm of Tarboton (1997) is single flow algorithm (see e.g. page 2, line 9, and page 3, line 22). Dinf rout water to one or two downslope cells. This is even shown on Figure 7 in the ms where the authors rightly show that Dinf often rout water to two downslope cells. Therefore it is misleading to call it single flow routing algorithm.

Overall, I think that the ms would benefit from moving from the dichotomy of SFD and MFD algorithms (especially given the fact that Dinf is not SFD algorithm). In my view, authors should compare and discuss the model results among all four algorithms. In the present version of the ms, authors usually state that results from SFM differ from MFD algorithm, but it is often unclear which particular algorithm authors really mean.

I do not understand why authors renamed well know RHESS model to CHESS (see page 3, line 3). As far as I can see from the text, these two models are the same. To use the different name for the same model is therefore misleading.

Figure 4: Figure caption is incomplete as there is no explanation what shows individual panels. Which panel is for D8 and which for MD8? Why authors showed only two of four algorithms compared in the ms?

Minor comments:

Suggestion for the title: I would recommend to replace "direction simulations based on" by "routing algorithms used in"

page 15, line 14: wrong formatting of the Reference Costa-Cabral & Burges

page 25, line 4: "A conceptual map ..." – I think that "A conceptual figure..." would be better

page 25, line 4: Figure caption is clearly not complete and something is missing at the end.

---

## Referee Comment (RC2) · Anonymous Referee #2 · 4 Apr 2018

SUMMARY:

This study compared single flow direction (SFD) vs. multiple flow direction (MFD) approaches in the derivation of DEM inputs for the distributed hydrologic model CHESS (coupled hydrology and ecology simulation system). The analysis was based on pixel-level and patch-level comparisons and spatial autocorrelation of ecohydrological variables, soil saturation deficit (SSD) and leaf area index (LAI) produced by the model with inputs from different flow routing schemes. Simulations covered a 12-year period, including one wet year and one dry year, in the semi-arid Cleve Creek water-

shed, Nevada, USA. To examine spatial autocorrelation of simulated variables, the study used Moran's I to relate mean SSD and LAI at the patch scale to Tesfa's (2011) distance-to-stream metric. SFD vs. MFD simulations produced similar streamflow values and similar watershed-scale mean values of SSD and LAI. SFD vs. MFD simulations produced different cell-level values of SSD and LAI, where differences were greatest in areas farthest from channels. In contrast, hydrologic variables were most different near channels. Spatial autocorrelation of SSD and LAI based on MFD was greater than that based on SFD, likely due to a higher degree of flow dispersion under MFD.

GENERAL COMMENTS:

The contribution of this paper lies in evaluation of model behavior rather than understanding of physical processes. In its current form, the value of the contribution of this paper is hard to discern, beyond the finding that differences between hydrograph simulations using the different routing methods evaluated were very small. The main differences reported among routing algorithms were in terms of autocorrelation of the ecohydrological variables (SSD and LAI) evaluated across grid cells. The conclusions state that ecohydrological variables are more autocorrelated under the MFD model, but do not say whether this is good, or why this is important. The paper does not establish why, or for what purpose, the degree of autocorrelation is a quantity of interest or how it relates to model performance. Are these autocorrelation quantities measures of how well the model performs? The contribution of the paper may be stronger if the authors are able to address this concern.

The paper is also unclear on how these findings might apply to future distributed hydrologic modeling research. Is it likely that the model behavior observed in this study will apply to other ecohydrologic models and other geographic regions? I hope that the authors can address this comment in the Discussion to increase the value of the paper's findings.

**HESSD**

The methods and conclusions of this paper would be much clearer if the authors would explicitly define their terminology pertaining to "flow direction", as it relates to SFD/MFD or SD/MD, in terms of not only possible flow directions but also in terms of the number of possible flow paths. Some of the studies cited within this paper have used "MFD" to describe a single flow path routed between 2 downslope cells, whereas other papers (including this one) use "MFD" and "MD" to describe multiple possible flow paths. Please clarify this terminology early in the paper. I suggest use SD for the single flow direction approach throughout, not SFD. Similarly use MD for multiple flow direction, and not MFD. For specific variations on MD, such as D-Infinity (Tarboton 1997) of MD-Infinity (Seibert and McGlynn 2007) just use these terms and mention that they are specific cases of MD.

Although this paper presents several descriptive statistics of SSD and LAI, including mean, range, min, max, and standard deviation, it would be more informative to add a figure showing the actual distribution of pixel-level SSD and LAI values as a histogram or density function. For example, probability density functions for SSD and LAI, for each flow routing algorithm and for differences between SD8 and MD8, would provide evidence of similarities or differences that may not be fully expressed by statistics. Boxplots would also be useful for showing the full distribution of SSD and LAI.

Previous papers (Tarboton 1997; Seibert and McGlynn 2007) described and demonstrated examples where over-dispersion of flow among multiple flow paths is unrealistic and thus undesirable. Please comment on how or why this is not a concern is your results, especially given that your results suggest that the spatial autocorrelation of eco-hydrological variables is greater under MFD than under SFD due to flow dispersion. This topic would fit nicely in section 4.2 of your Discussion.

Some of the analytical methods used in this paper were not described in sufficient detail to evaluate your choice of methods. Specifically, please address or clarify: 1) exactly how CHESS differs from RHESSys; 2) the extent to which CHESS was calibrated individually with each routing algorithm; 3) differences between the MD-infinity

and RMD-infinity algorithms; 4) categories used to classify distance-from-stream, i.e., to convert continuous numeric to categorical variables; 5) method for delineation of patches; 6) identification of "wet" and "dry" years; and 7) the specific tests used to assess statistical significance of differences in Nash-Sutcliffe efficiencies. Items #4 and #5 are particularly critical for evaluating your results because wider numerical categories will include a greater range of values in the same category, and thus have a higher kappa, relative to smaller and thus more precise categories. See below (specific comments) for additional feedback and suggestions pertaining to these specific items.

SPECIFIC COMMENTS:

Page 1, lines 8-22: It would be helpful for the abstract to name the distributed hydrologic model (CHESS) and ecohydrological variables used in the comparisons (SSD and LAI).

Page 2, lines 7-8: Specifically, flow follows the direction of steepest downwards topographic slope (which is more specific than "...follows the topographic relief").

Page 3, lines 4-10: Briefly clarify how CHESS is different from RHESSys. The statement that "specific algorithms for carbon, water, and nutrient dynamics... are mostly maintained as in Tague and Band (2004)" is confusing and requires further explanation. What is "mostly"? Tague and Band (2004) indicate that RHESSys relies on either TOP-MODEL or DHSVM for routing. How is CHESS different, and which (if any) aspects of RHESSys's routing algorithms are retained in your simulations?

Page 3, line 29: Beginning with this paragraph, for clarity please explicitly state which routing algorithm is being discussed in each paragraph.

Page 4: Given that the methods used (D8, D-infinity, MD8 and MD-infinity) have all been described in detail in the publications cited in this paper, the equations and methods do not need to be presented in as much detail as they are presented here. One exception is MD-infinity, which should be clearly described in terms of its difference from RMD-infinity.

[Figure]

Page 4, line 16: Where the citation is provided for MD-infinity, it should also be provided for D-infinity.

Page 4, line 19: Briefly summarize what you mean by "...the advantages of D-infinity and MD8".

Page 4, lines 16-24: The reason for the adoption of a new method (RMD-infinity) is not clear. How does this improve upon MD-infinity, and how can you quantify this? It seems that dividing flow among all triangular facets reintroduces the problem of unrealistic dispersion on convergent slopes, as described in Tarboton (1997) and Seibert and McGlynn (2007). RMD-Infinity is a new terrain flow routing approach. It does not do justice to it as a potential contribution to flow routing methodology to introduce it without presenting a more detailed evaluation and conclusion as to its efficacy.

Page 5, line 6: Explain how the land cover data "are pre-specified". What is the source?

Page 5, lines 15-18: This section states that calibration was done for each of the four routing algorithms, while the following statement indicates that model parameterizations were almost identical among the four simulations. These statements appear contradictory. Please clarify how the calibration methods accounted for any streamflow differences among the four simulations. Also consider that Wolock and McCabe (1995) found that separate calibration for models using alternative routing methods affected accuracy of simulated streamflow, and discuss how your findings compare with their findings in your Discussion.

Page 5, lines 24-25: Means and standard deviations are only two metrics that can describe a distribution, and they are often inadequate at detecting important differences among multiple distributions. Please also consider showing the entire distribution in the form of a probability density function, histogram or boxplot.

Page 6, lines 12-14: Neither citations for these metrics nor the methods used to delineate patches are presented here. Please specify the numerical categories that were

used to classify distance-from-stream. Also describe how patches were delineated, as well as their number and range of sizes.

Page 7, line 2: This is the first mention (other than in the Abstract) of "wet year" or "dry year". In Methods, describe how "wet" and "dry" years were identified, with at least minimal data to support the identification of these years.

Page 7, line 28: Was an actual significance test applied to the NS values? If yes, what test (describe in Methods)? If no, this sentence should describe differences as small rather than "no significant difference".

Page 8, line 15: What were the pre-specified ranges of values? These should be stated in Methods.

Page 10, lines 6-11: Radula et al. (2018) also compared differences in simulated soil moisture among several flow routing algorithms. They evaluated regressions between soil moisture and topographic wetness index, and also between ecological indicators of soil moisture and wetness index, where wetness index was estimated under different flow routing algorithms. I suggest comparing your findings to theirs in the Discussion.

Figure 1: Please specify the source of the land cover information shown in the map.

Figure 4: As described in the Methods, this analysis uses means within patches. This detail should be specified in the caption; otherwise it implies that distance-from-channel for individual pixels were used.

Figure 7: This figure does not appear to present any new information beyond what Seibert and McGlynn (2007) showed. I suggest eliminating this figure (or alternatively, clarifying how it expands on previous work).

CITATIONS:

Raduła, M. W., Szymura, T. H., & Szymura, M. (2018). Topographic wetness index explains soil moisture better than bioindication with Ellenberg's indicator values. Ecological Indicators, 85, 172-179.

Seibert, J., & McGlynn, B. L. (2007). A new triangular multiple flow direction algorithm for computing upslope areas from gridded digital elevation models. Water resources research, 43(4).

Tarboton, D. G. (1997). A new method for the determination of flow directions and upslope areas in grid digital elevation models. Water resources research, 33(2), 309-319.
* * *

---

## Author Comment (AC2) · 3 May 2018

Dear Referee #2: We greatly appreciate your valuable comments on our manuscript (hess-2018-47). We have carefully addressed all of your comments and our responses are listed below one by one following each of your comments!

General Comments:

(1)The methods and conclusions of this paper would be much clearer if the authors would explicitly define their terminology pertaining to "flow direction", as it relates to

SFD/MFD or SD/MD, in terms of not only possible flow directions but also in terms of the number of possible flow paths. Some of the studies cited within this paper have used "MFD" to describe a single flow path routed between 2 downslope cells, whereas other papers (including this one) use "MFD" and "MD" to describe multiple possible flow paths. Please clarify this terminology early in the paper. I suggest use SD for the single flow direction approach throughout, not SFD. Similarly use MD for multiple flow direction, and not MFD. For specific variations on MD, such as D-Infinity (Tarboton 1997) of MDInfinity (Seibert and McGlynn 2007) just use these terms and mention that they are specific cases of MD.

Response:

Thanks for your good comments! We decide to define the terminology pertaining to "flow direction" in terms of the maximum number of possible flow paths to adjacent downslope cells. Accordingly, we used "SD" instead of "SFD" and "MD" instead of "MFD" to describe these flow routing algorithms. Thus, D-Infinity is defined as a special case of MD algorithms since it allows two flow paths to downslope cells. We have added related definitions in the "Introduction" section of the revised manuscript. These terminologies are kept consistent throughout our manuscript.

In addition, we added one more flow routing algorithm, i.e., MFD-md, to the revised manuscript. As a result, a total of five algorithms (i.e., D8, D-Infinity, RMD-Infinity, MD8, MFD-md) were used in the revised manuscript and results among the five routing algorithms are compared, respectively. For example, we treated D-Infinity as a specific case of MD algorithms. We then compared the ecohydrological variables SSD and LAI between each pair of five algorithms at cell level as we did before in the Section 3.4 and 3.5 for the year 1992 (a relatively dry year) and 2005 (a relatively wet year) as well as for the whole study period 1991-2012 (see Table S4 that will be added to the revised paper), respectively. Specially, four pairs of algorithms (i.e., D8/RMD_inf, D_inf/RMD_inf, D8/MD8, RMD_inf/MFD-md) are selected for comparative analysis in section 3.4 and 3.5 (Fig. S2 on the supplement of this reply). Correspondingly, Fig. 7

and Fig. 8 show comparisons of four pairs of algorithms (Fig. S3). For Fig. 9, we added the relationship between leaf area index (LAI) and "distance to stream (a, b)" as well as relationship between SSD and LAI (c) under five algorithms to offer more details. We believe that these changes in the selection of algorithms and data comparison will help readers to understand our results.

(2)Although this paper presents several descriptive statistics of SSD and LAI, including mean, range, min, max, and standard deviation, it would be more informative to add a figure showing the actual distribution of pixel-level SSD and LAI values as a histogram or density function. For example, probability density functions for SSD and LAI, for each flow routing algorithm and for differences between SD8 and MD8, would provide evidence of similarities or differences that may not be fully expressed by statistics. Boxplots would also be useful for showing the full distribution of SSD and LAI.

Response:

Thanks for your good comments! An additional figure (Figure S1) including both boxplot and density plot are added in supplement and will be eventually added/cited in Section 3.2 in the revised manuscript.

(3)The conclusions state that ecohydrological variables are more autocorrelated under the MFD model, but do not say whether this is good, or why this is important. The paper does not establish why, or for what purpose, the degree of autocorrelation is a quantity of interest or how it relates to model performance. Are these autocorrelation quantities measures of how well the model performs? The contribution of the paper may be stronger if the authors are able to address this concern.

Response:

Thanks! Theoretically, the Moran's I quantifies the degree of similarities of nearby locations in space. It measures the spatial autocorrelation of a variable and expresses the homogeneity of a variable in its neighborhood extent. In this study, we refer the

major differences between Moran's I (similarity between nearby cells) to their differentiated degrees in "dispersion" (extent of flow divergence) of model results. We used this index to not only provide statistical evidence and but also quantify the extent of "dispersion" that can not be measured by standard deviation (Fig. S2, difference < 2.8 % in SSD). Overall, according to the values of Moran's I (Fig. S3, difference ranges from 3% to 26% for SSD), we think differences between model results in terms of "dispersion" do exist for some distinct algorithms (e.g. D8 vs. MD8) but the extent of dispersion is not as significant as some previous studies suggested partially because they often illustrate cases in some drainage zones by theoretical DEMs (e.g. divergent hillslopes). We have added related information in the revised manuscript.

(4)Previous papers (Tarboton 1997; Seibert and McGlynn 2007) described and demonstrated examples where over-dispersion of flow among multiple flow paths is unrealistic and thus undesirable. Please comment on how or why this is not a concern is your results, especially given that your results suggest that the spatial autocorrelation of eco-hydrological variables is greater under MFD than under SFD due to flow dispersion.

Response:

Thanks! This is a very challenging question. From my point of view, it's hard to judge whether "dispersion" is good or not without field data and whether some routing algorithms with maximum number of flow paths (e.g. possibly 8 paths in MD8, MFD-md) can results in an "over-dispersion" issue. It might be seen as a theoretical advantages for D-Infinity over MD-Infinity. However, previous studies that were based field observations (e.g., Kopecká and Čížková , 2010; Tang et al., 2014; Radula et al., 2018) indicated that MD algorithms with greater possible flow paths (e.g., MD8, MFD-md, TFM) tended to provide better results. We have discussed this in the revised manuscript.

(5)This topic would fit nicely in section 4.2 of your Discussion. Some of the analytical methods used in this paper were not described in sufficient detail to evaluate your choice of methods. Specifically, please address or clarify: 1)exactly how CHESS differs

from RHESSys; 2) the extent to which CHESS was calibrated individually with each routing algorithm; 3) differences between the MD-infinity and RMD-infinity algorithms; 4) categories used to classify distance-from-stream, i.e., to convert continuous numeric to categorical variables; 5) method for delineation of patches; 6) identification of "wet" and "dry" years; and 7) the specific tests used to assess statistical significance of differences in Nash-Sutcliffe efficiencies. Items #4 and #5 are particularly critical for evaluating your results because wider numerical categories will include a greater range of values in the same category, and thus have a higher kappa, relative to smaller and thus more precise categories. See below (specific comments) for additional feedback and suggestions pertaining to these specific items.

Response:

**1 A detailed description about how CHESS differ from RHESSys is given in reply on Specific Comments #3.**

**2 Additional soil text parameterizations for model's calibration are given in supplement of this response and will be finally put in the revised manuscript (Table S1).**

**3 RMD-infinity allows all triangular facets with possible flow paths to participate in flow partitioning i.e. all "routing of flow" while MD-infinity considers all "routing of area" which allows one facet of each area participate in flow partitioning on convex hillslopes. Specifically, in cell's numbers of flow paths (Fig. S5), MD-infinity is more similar to D-infinity and RMD-infinity is more similar to MD8 but they mainly differ in the portion of flow in each flow path. When comparing D-infinity with RMD-infinity, the results from RMD-Inifinty are more similar to those of MD8 and MFD-md than are MD-infinity while both are based on triangular facets in their calculations (Fig. S5, Table S4).**

**4 the "Distance-to-stream" is calculated as an integer number for every cell and each integer number stands for a specific type. Since the distances for most cells to their nearest stream-type cells are not at the cardinal direction (E, W, N, S), the distance will be calculated as a specific integer but not numbers like 100, 500. So there are**

more and more types as distance rises (e.g., 100, 141(100$\sqrt{2}$), 200, 223(100$\sqrt{5}$),...) as in Fig.4. We didn't turn it into categorical numbers like 100, 200, 300 because fewer points is usually provided with higher R2 . We want to remain more details in calculating D_R while most distances with specific integer number have more than 10 cells.

**5 Because the CHESS model used in this study is process-based distributed model, the patches (or cells) are delineated by soil, vegetation and topographic features including slopes, aspects and elevations.**

**6 The relatively wet and dry year are determined by the amount of annual total rainfall in a water-year (will furtherly given in our new supplement, Table S1). We have added related information in the revised manuscript.**

**7 We accepted your advice and use "small" rather than "no significant" difference in NS.**

Specific Comments:

(1)Page 1, lines 8-22: It would be helpful for the abstract to name the distributed hydrologic model (CHESS) and ecohydrological variables used in the comparisons (SSD and LAI).

Response:

Thanks for your good comments. We have added these information in the abstract section.

(2)Page 2, lines 7-8: Specifically, flow follows the direction of steepest downwards topographic slope (which is more specific than ": : :follows the topographic relief").

Response:

We accepted your good suggestion and have revised it.

(3)Page 3, lines 4-10: Briefly clarify how CHESS is different from RHESSys. The statement that "specific algorithms for carbon, water, and nutrient dynamics: : : are mostly maintained as in Tague and Band (2004)" is confusing and requires further explanation. What is "mostly"? Tague and Band (2004) indicate that RHESSys relies on either TOPMODEL or DHSVM for routing. How is CHESS different, and which (if any) aspects of RHESSys's routing algorithms are retained in your simulations?

Response:

Thanks for your good comments! Actually, we renamed "R-RHESSys (note: instead of RHESSys)" to CHESS, which is short for "Coupled Hydrology and Ecology Simulation Systems". Tang et al. (2014, 2016) developed R-RHESSys based on RHESSys modelãĂĆAs discussed in Tang et al. (2014), we have removed the hierarchical structure of the original RHESSys model and also excluded the top-model embedded in the original RHESSys. In addition, we have redesigned the model-user interface for R-RHESSys and modified model codes much. We renamed "R-RHESSys" to CHESS for the purpose of its future development and usage. We have revised relevant text in the revised manuscript for clarification The explicit and implicit routing approaches used in this study represent two basic approaches to model lateral soil moisture flux (Tague and Band, 2001). Spatial variability in soil moisture also can be addressed implicitly using statistical distribution methods such as TOPMODEL (Beven and Kirkby, 1979), which distribute saturation deficit across a non-spatial distribution of hydrological parameters within a catchment. Explicit routing approaches have been applied in models such as TOPOG (O'Loughlin, 1990), VSAS (Bernier, 1985), CLAWS (Duan, 1996) and DHSVM (Wigmosta et al., 1994), which explicitly transfer water between connected cells. R-RHESSys (Tang et al., 2014), excluded the TOPMODEL (Beven and Kirkby, 1979) embedded in the model's predecessor and retained the explicit water-routing algorithm. Thus, the major differences between our studies and Wolock and McCabe (1995) used TOPMODEL with implicit routing are that the former performed in a fully distributed mode with explicit routing. Besides, although many studies studied al-

gorithm's performance via estimation of theoretical results such as TWI, we are not certain whether flow routing algorithms can affect model's performance. Comparison between the two routing approaches has been well discussed and illustrates the advantage of the explicit routing approach, though the loss of computational efficiency associated with the explicit routing approach is noted (Tague and Band, 2001). So, we focus on comparisons between model results via explicit flow routing algorithms used in CHESS, which provides not only evidences on the effects of routing algorithms on model's performance but also insights into future efforts to improve model-based research in studying ecohydrological processes in the land surface. Briefly, compared to results of Wolock and McCabe (1995) using TOPMODEL, we found that an ecohydrological model with differing explicit routing algorithms, can be easily calibrated and parameterized to be able to simulate stream hydrograph with similar pattern. We will enroll these discussions in our revised manuscript.

(4)Page 3, line 29: Beginning with this paragraph, for clarity please explicitly state which routing algorithm is being discussed in each paragraph.

(5)Page 4: Given that the methods used (D8, D-infinity, MD8 and MD-infinity) have all been described in detail in the publications cited in this paper, the equations and methods do not need to be presented in as much detail as they are presented here. One exception is MD-infinity, which should be clearly described in terms of its difference from RMD-infinity.

Response to (4) and (5):

We will revised relevant parts for clarification. We retained descriptions of RMD-Infinity and MD-infinity. Results from other algorithms will be summarized with concise words.

(6)Page 4, line 16: Where the citation is provided for MD-infinity, it should also be provided for D-infinity.

Response:

[Figure]

We provided a citation for D-Infinity.

(7)Page 4, line 19: Briefly summarize what you mean by ": : :the advantages of D-infinity and MD8".

(8)Page 4, lines 16-24: The reason for the adoption of a new method (RMD-infinity) is not clear. How does this improve upon MD-infinity, and how can you quantify this? It seems that dividing flow among all triangular facets reintroduces the problem of unrealistic dispersion on convergent slopes, as described in Tarboton (1997) and Seibert and McGlynn (2007). RMD-Infinity is a new terrain flow routing approach. It does not do justice to it as a potential contribution to flow routing methodology to introduce it without presenting a more detailed evaluation and conclusion as to its efficacy.

Response to (7) and (8):

There are two theoretical advantages for D-Infinity: (i) First, D-infinity can more accurately describe water routing than does D8 by assuming infinite possible flow routing in a triangular facet based on DEMs; (ii) Second, it limits the maximum number of flow paths to 2 downslope cells to avoid "dispersion". However, more and more field studies have shown a favor of MD algorithms such as MD8. And since differences between D-Infinity and MD-Infinity is minimal in our studies (6.7% in cells' numbers of flow paths) though Cleve Creek is a mountainous watershed with great topographic relief. To discuss the extent of differences between D-Infinity and MD-Infinity (resolution of DEM) is somewhat beyond the scope of our current studies. Nevertheless, we will explore such issue in the future study.

Besides, we tested if RMD-Infinity is different from other MD algorithms (MD8, MFD-md) in model's results as we have done throughout the studies (Fig.S2 & Table. S4). The results indicate that RMD-infinity differs to some degree from other algorithms. This might offer additional insights into future ecohydrological modeling studies.

(9)Page 5, line 6: Explain how the land cover data "are pre-specified". What is the
source?

Response:

The land cover data used in this study were provided by the courtesy of Dr. Lutz when Dr. Tang conduct the US DOE research project, for which Dr. Lutz is the PI. We have added related information in the proper place.

(10)Page 5, lines 15-18: This section states that calibration was done for each of the four routing algorithms, while the following statement indicates that model parameterizations were almost identical among the four simulations. These statements appear contradictory. Please clarify how the calibration methods accounted for any streamflow differences among the four simulations. Also consider that Wolock and McCabe (1995) found that separate calibration for models using alternative routing methods affected accuracy of simulated streamflow, and discuss how your findings compare with their findings in your Discussion.

Response:

Actually, we only slightly modify the values of soil hydraulic conductivity decay parameter through which we can have model to simulate well the observed streamflow under all different algorithms. The differentiated parameters for calibration hydraulic conductivity with depth (m) is given in Table S1 along with other undifferentiated parameters of soil texture. Thanks for offering another point to explain our results and we will revise the discussion Section 4.1 as well as referring to studies of Wolock and McCabe (1995) .

(11)Page 5, lines 24-25: Means and standard deviations are only two metrics that can describe a distribution, and they are often inadequate at detecting important differences among multiple distributions. Please also consider showing the entire distribution in the form of a probability density function, histogram or boxplot.

Response:

Thanks, we have used other plots such as boxplot and density plots in the revised manuscript. Also, see our response to the General Comment (3).

(12)Page 6, lines 12-14: Neither citations for these metrics nor the methods used to delineate patches are presented here. Please specify the numerical categories that were used to classify distance-from-stream. Also describe how patches were delineated, as well as their number and range of sizes.

Response:

Thanks, please see our response to the General Comment (4)#4.

(13)Page 7, line 2: This is the first mention (other than in the Abstract) of "wet year" or "dry year". In Methods, describe how "wet" and "dry" years were identified, with at least minimal data to support the identification of these years.

Response:

It is determined by the amount of annual total rainfall. We will put related information in the revised manuscript. Table 1.).

(14)Page 7, line 28: Was an actual significance test applied to the NS values? If yes, what test (describe in Methods)? If no, this sentence should describe differences as small rather than "no significant difference".

Response:

We accept your advice and use "small" rather than "no significant" to describe the differences in NS.

(15)Page 8, line 15: What were the pre-specified ranges of values? These should be stated in Methods.

Response:

As in our response to the General Comments(4)#5, the six ranges are the same as in

Fig. 3 and Fig.5 to calculated the Kappa statistics (For SSD and LAI).

(16)Page 10, lines 6-11: Radula et al. (2018) also compared differences in simulated soil moisture among several flow routing algorithms. They evaluated regressions between soil moisture and topographic wetness index, and also between ecological indicators of soil moisture and wetness index, where wetness index was estimated under different flow routing algorithms. I suggest comparing your findings to theirs in the Discussion.

Response:

In our model, the baseflow generated form a cell is determined by the transmissivity curve and via explicit routing algorithm. Thus, we considered to discuss the differences in model results between TWI and explicit routing algorithms in relevant discussion section in our revised manuscript.

(17)Figure 1: Please specify the source of the land cover information shown in the map.

Response:

See our response to the Specific Comment (9) above.

(18)Figure 4: As described in the Methods, this analysis uses means within patches. This detail should be specified in the caption; otherwise it implies that distance-from-channel for individual pixels were used.

Response:

Thanks and we have added these information in our new figure caption (Fig. S2).

(19)Figure 7: This figure does not appear to present any new information beyond what Seibert and McGlynn (2007) showed. I suggest eliminating this figure (or alternatively, clarifying how it expands on previous work).

Response:

Thanks and we removed it and now kept it in Fig. S2 of the revised supplement.

Again, we greatly appreciate your valuable comments that have helped greatly improve our manuscript.

Yours Sincerely,

Zhenwu Xu

2018/5/3

Major References

Raduła, M. W., Szymura, T. H., and Szymura, M.: Topographic wetness index explains soil moisture better than bioindication with Ellenberg's indicator values, Ecological Indicators, 85, 172-179, https://doi.org/10.1016/j.ecolind.2017.10.011, 2018.

Tague, C. L., and Band, L. E.: Evaluating explicit and implicit routing for watershed hydro-ecological models of forest hydrology at the small catchment scale, Hydrological Processes, 15, 1415-1439, 10.1002/hyp.171, 2001.

Tang, G., Hwang, T., and Pradhanang, S. M.: Does consideration of water routing affect simulated water and carbon dynamics in terrestrial ecosystems?, Hydrology and Earth System Sciences, 18, 1423-1437, 10.5194/hess-18-1423-2014, 2014.

Wolock, D. M., and McCabe, G. J.: Comparison of single and multiple flow direction algorithms for computing topographic parameters in TOPMODEL, Water Resources Research, 31, 1315–1324, 10.1029/95WR00471, 1995.

Please also note the supplement to this comment:
https://www.hydrol-earth-syst-sci-discuss.net/hess-2018-47/hess-2018-47-AC2-supplement.pdf

[Figure]

**Supplement:**

[Figure]

**Figure S1.** The boxplot and density plot of average soil saturation deficit and leaf area index among the five routing algorithms during 1991-2012. The oblique cross-shaped patterns in (a) and (b) are mean values for each data set.

[Figure]

**Figure S2.** Changes in the averaged relative deviation ($D_{AR}$) of soil water saturation deficit (SSD) and leaf area index (LAI) as distances from channel increase under four pairs of compared algorithms for the 1992-dry and 2005-wet year. $D_{AR}$ are averaged $D_R$ for each distance (integer number) by Eq. 9. The p-value is less than 0.01 in all linear regression models.

[Figure]

**Figure S3.** (a) Changes in relative deviations ($D_R$) of stream-type cells in accumulated area of flow under different algorithms' simulations decrease apparently as flow moves from the head to the outlet of the watershed and (b) Changes in the average relative deviation ($D_{AR}$) of accumulated area of flow between simulations increases significantly as the distance of cells to channels increases under four compared algorithms. the p-value is less than 0.01 in all linear regression models.

[Figure]

**Figure S4.** Relationships of average soil saturation deficit (SSD) and leaf area index (LAI) to "distance to stream" (a, b) as well as relationship between SSD and LAI (c) under five routing algorithms.

[Figure]

**Figure S5.** The total number of cells where flow is routed to from 1 to 8 downslope neighbors. The digits in x-axis refers to the number of flow paths to downslope neighbors from a center cell.

**Table S1.** Parameterizations of major soil parameters used in the model simulations.

| Variables | | Unit | Soil texture | | |
|---|---|---|---|---|---|
| | | | Gravelty loam | Fine sandy loam | Fine sandy |
| | D8 | | 0.235 | 0.205 | 0.195 |
| | D∞ | | 0.232 | 0.202 | 0.192 |
| m* | RMD∞ | DIM | 0.240 | 0.210 | 0.200 |
| | MD8 | | 0.246 | 0.216 | 0.206 |
| | MFD-md | | 0.244 | 0.214 | 0.204 |
| $K_{sat_0}$* | | m day$^{-1}$ | 128.36 | 109.56 | 132.73 |
| Porosity | | % | 0.451 | 0.475 | 0.485 |
| Soil depth | | m | 4.8 | 5.0 | 5.2 |
| Active zone depth | | m | 10.0 | 10.0 | 10.0 |
| Albedo | | DIM | 0.32 | 0.20 | 0.37 |
| Sand | | % | 0.68 | 0.60 | 0.82 |
| Clay | | % | 0.15 | 0.22 | 0.10 |
| Silt | | % | 0.17 | 0.18 | 0.8 |

*m is the decay rate of hydraulic conductivity with depth. $K_{sat_0}$ is saturated hydraulic conductivity at the surface; m were manually calibrated against observed streamflow and derived baseflow at the USGS gauge station under five algorithms respectively.

**Table S2.** Comparisons of modeled soil saturation deficit and leaf area index among the four routing algorithms averaged for the watershed.

| | Soil saturation deficit | | | | | Leaf area index | | | | |
|---|---|---|---|---|---|---|---|---|---|---|
| | D8 | D∞ | RMD∞ | MD8 | MFD-md | D8 | D∞ | RMD∞ | MD8 | MFD-md |
| Min | 0.058 | 0.026 | 0.036 | 0.054 | 0.038 | 0.000 | 0.000 | 0.000 | 0.000 | 0.000 |
| Max | 1.856 | 1.840 | 1.879 | 1.905 | 1.882 | 1.297 | 1.298 | 1.299 | 1.298 | 1.296 |
| Mean | 1.387 | 1.389 | 1.384 | 1.383 | 1.369 | 0.294 | 0.285 | 0.303 | 0.302 | 0.314 |
| σ | 0.290 | 0.290 | 0.285 | 0.283 | 0.281 | 0.265 | 0.253 | 0.260 | 0.267 | 0.271 |

*Statistics are calculated based on mean daily values averaged for the study period 1991-2012 at cell level.

**Table S3.** Comparisons of the spatial autocorrelation (measured by Moran's I) of modeled values among the five routing algorithms

| Algorithms | Soil saturation deficit | | | Leaf area index | | |
|---|---|---|---|---|---|---|
| | 1992 | 2005 | 1991-2012 | 1992 | 2005 | 1991-2012 |
| D8 | 0.418 | 0.414 | 0.425 | 0.417 | 0.436 | 0.424 |
| D∞ | 0.433 | 0.425 | 0.436 | 0.414 | 0.431 | 0.419 |
| RMD∞ | 0.487 | 0.479 | 0.494 | 0.445 | 0.462 | 0.451 |
| MD8 | 0.507 | 0.500 | 0.515 | 0.456 | 0.474 | 0.463 |
| MFD-md | 0.528 | 0.516 | 0.535 | 0.462 | 0.483 | 0.467 |
| p | < 0.01 | | | | | |

**Table S4.** Comparisons of cell-level $D_R$ averaged for the watershed between compared algorithms

| Algorithms compared | | Soil saturation deficit | | | Leaf area index | | |
|---|---|---|---|---|---|---|---|
| | | 1992 | 2005 | 1991-2012 | 1992 | 2005 | 1991-2012 |
| D8 | D∞ | 0.063 | 0.068 | 0.065 | 0.181 | 0.184 | 0.178 |
| D8 | RMD∞ | 0.062 | 0.067 | 0.064 | 0.174 | 0.174 | 0.177 |
| D8 | MD8 | 0.062 | 0.067 | 0.063 | 0.175 | 0.175 | 0.175 |
| D8 | MFD-md | 0.065 | 0.070 | 0.067 | 0.169 | 0.172 | 0.176 |
| D∞ | RMD∞ | 0.027 | 0.029 | 0.028 | 0.114 | 0.116 | 0.108 |
| D∞ | MD8 | 0.034 | 0.037 | 0.035 | 0.132 | 0.134 | 0.124 |
| D∞ | MFD-md | 0.043 | 0.046 | 0.044 | 0.154 | 0.156 | 0.147 |
| RMD∞ | MD8 | 0.014 | 0.015 | 0.014 | 0.061 | 0.062 | 0.054 |
| RMD∞ | MFD-md | 0.026 | 0.028 | 0.026 | 0.098 | 0.098 | 0.089 |
| MD8 | MFD-md | 0.020 | 0.022 | 0.021 | 0.080 | 0.081 | 0.072 |

---

## Author Comment (AC3) · 3 May 2018

Dear Referee #1:

We greatly appreciate your valuable comments on our manuscript (#hess-2018-47). We have carefully addressed all of your comments and our responses are listed below one by one following each of your comments!

Specific Comments:

[Figure]

(1)I do not agree with the authors that Dinf algorithm of Tarboton (1997) is single flow algorithm (see e.g. page 2, line 9, and page 3, line 22). Dinf rout water to one or two downslope cells. This is even shown on Figure 7 in the ms where the authors rightly show that Dinf often rout water to two downslope cells. Therefore it is misleading to call it single flow routing algorithm.

Response:

Thanks for your good comments! Yes, D_inf is an specific case of MFD algorithms. Following Referee #2 's advices, we used "SD" instead of "SFD" and "MD" instead of "MFD" to describe these routing algorithms. The definition will be added in Introduction section of the revised manuscript. These terminologies are consistent throughout our manuscript. Besides, we have considered the advices from Dr. Qin (SC) to add a new algorithm MFD-md to the revised manuscript and now a total of five algorithms are kept (D8, D-Infinity, MD8, RMD-Infinity, MFD-md) in the revised manuscript.

(2)Overall, I think that the ms would benefit from moving from the dichotomy of SFD and MFD algorithms (especially given the fact that Dinf is not SFD algorithm). In my view, authors should compare and discuss the model results among all four algorithms. In the present version of the ms, authors usually state that results from SFD differ from MFD algorithm, but it is often unclear which particular algorithm authors really mean.

Response:

Thanks for your good comments! The dichotomy of SFD and MFD algorithms are now removed from the title and we only refer to SD and MD to distinguish their differences in dispersion of modeled data (see Section 3,4, 4.2 in the revised manuscript) while we define D_Inf is a case of MD under special circumstance.

Specially, four pairs of algorithms (D8/RMD_inf, D_inf/RMD_inf, D8/MD8, RMD_inf/MFD-md) are selected for comparisons in section 3.4 and 3.5 (Fig. S2 in the supplement of this reply). The cell-level DR averaged for the watershed ranges

from 2.6% to 6.4% under these four representative pairs of algorithms (Table S4.). Correspondingly, more details have been added to Fig. 7 and Fig 8 to demonstrate the comparisons between the four paired algorithms (see Fig. S3 in supplementary materials). For Fig. 9, we added the relationships between leaf area index (LAI) and "distance to stream" (panel a, b) as well as relationship between SSD and LAI (c) under five algorithms. Nevertheless, an representative example of D8 vs RMD_inf will be remained in our revised manuscript for distribution of SSD and LAI in Fig. 3 and Fig. 5 because it's hard and also redundant to show all groups of compared algorithms. We believe that these changes in the selection of algorithms and comparisons of model results as well as consequent Figures and Tables will offer more details for readers to understand our results.

(3)I do not understand why authors renamed well know RHESS model to CHESS (see page 3, line 3). As far as I can see from the text, these two models are the same. To use the different name for the same model is therefore misleading.

Response:

Thanks for your good comments! Actually, we renamed "R-RHESSys (note: instead of RHESSys)" to CHESS, which is short for "Coupled Hydrology and Ecology Simulation Systems". Tang et al. (2014, 2016) developed R-RHESSys based on RHESSys modelãĂĆAs discussed in Tang et al. (2014), we have removed the hierarchical structure of the original RHESSys model and also excluded the top-model embedded in the original RHESSys. In addition, we have redesigned the model-user interface for R-RHESSys and modified model codes much. We renamed "R-RHESSys" to CHESS for the purpose of its future development and usage. We have revised relevant text in the revised manuscript for clarification.

(4)Figure 4: Figure caption is incomplete as there is no explanation what shows individual panels. Which panel is for D8 and which for MD8? Why authors showed only two of four algorithms compared in the ms?

[Figure]

Response: Thanks for your good comments! As our responses to the Specific Comments (2), four pairs of algorithms are compared, respectively, in the revised manuscript.

Minor comments:

(1)Suggestion for the title: I would recommend to replace "direction simulations based on by "routing algorithms used in".

Response:

Thanks for your advices. The title has been changed to "Similarity and dissimilarity in model-results among flow routing algorithms used in a distributed ecohydrological model"

(2)page 15, line 14: wrong formatting of the Reference Costa-Cabral & Burges

Response:

We have revised it.

(3)page 25, line 4: "A conceptual map : : :" – I think that "A conceptual figure: : :" would be better

Response:

Thanks for your good comments! We changed it to "a conceptual figure".

(4)page 25, line 4: Figure caption is clearly not complete and something is missing at the end.

Response:

Thanks for your good comments! We revised relevant figure captions in our revised manuscript .

Overall, we benefited much from your comments and hope that our responses to each
of your comments are satisfactory. Again, we greatly appreciate your valuable comments.

Yours Sincerely,

Zhenwu Xu

2018/5/3

Please also note the supplement to this comment:
https://www.hydrol-earth-syst-sci-discuss.net/hess-2018-47/hess-2018-47-AC3-supplement.pdf

**Supplement:**

[Figure]

**Figure S1.** The boxplot and density plot of average soil saturation deficit and leaf area index among the five routing algorithms during 1991-2012. The oblique cross-shaped patterns in (a) and (b) are mean values for each data set.

[Figure]

**Figure S2.** Changes in the averaged relative deviation ($D_{AR}$) of soil water saturation deficit (SSD) and leaf area index (LAI) as distances from channel increase under four pairs of compared algorithms for the 1992-dry and 2005-wet year. $D_{AR}$ are averaged $D_R$ for each distance (integer number) by Eq. 9. The p-value is less than 0.01 in all linear regression models.

[Figure]

**Figure S3.** (a) Changes in relative deviations ($D_R$) of stream-type cells in accumulated area of flow under different algorithms' simulations decrease apparently as flow moves from the head to the outlet of the watershed and (b) Changes in the average relative deviation ($D_{AR}$) of accumulated area of flow between simulations increases significantly as the distance of cells to channels increases under four compared algorithms. the p-value is less than 0.01 in all linear regression models.

[Figure]

**Figure S4.** Relationships of average soil saturation deficit (SSD) and leaf area index (LAI) to "distance to stream" (a, b) as well as relationship between SSD and LAI (c) under five routing algorithms.

[Figure]

**Figure S5.** The total number of cells where flow is routed to from 1 to 8 downslope neighbors. The digits in x-axis refers to the number of flow paths to downslope neighbors from a center cell.

**Table S1.** Parameterizations of major soil parameters used in the model simulations.

| Variables | | Unit | Soil texture | | |
|---|---|---|---|---|---|
| | | | Gravelty loam | Fine sandy loam | Fine sandy |
| | D8 | | 0.235 | 0.205 | 0.195 |
| | D∞ | | 0.232 | 0.202 | 0.192 |
| m* | RMD∞ | DIM | 0.240 | 0.210 | 0.200 |
| | MD8 | | 0.246 | 0.216 | 0.206 |
| | MFD-md | | 0.244 | 0.214 | 0.204 |
| $K_{sat_0}$* | | m day$^{-1}$ | 128.36 | 109.56 | 132.73 |
| Porosity | | % | 0.451 | 0.475 | 0.485 |
| Soil depth | | m | 4.8 | 5.0 | 5.2 |
| Active zone depth | | m | 10.0 | 10.0 | 10.0 |
| Albedo | | DIM | 0.32 | 0.20 | 0.37 |
| Sand | | % | 0.68 | 0.60 | 0.82 |
| Clay | | % | 0.15 | 0.22 | 0.10 |
| Silt | | % | 0.17 | 0.18 | 0.8 |

*m is the decay rate of hydraulic conductivity with depth. $K_{sat_0}$ is saturated hydraulic conductivity at the surface; m were manually calibrated against observed streamflow and derived baseflow at the USGS gauge station under five algorithms respectively.

**Table S2.** Comparisons of modeled soil saturation deficit and leaf area index among the four routing algorithms averaged for the watershed.

| | Soil saturation deficit | | | | | Leaf area index | | | | |
|---|---|---|---|---|---|---|---|---|---|---|
| | D8 | D∞ | RMD∞ | MD8 | MFD-md | D8 | D∞ | RMD∞ | MD8 | MFD-md |
| Min | 0.058 | 0.026 | 0.036 | 0.054 | 0.038 | 0.000 | 0.000 | 0.000 | 0.000 | 0.000 |
| Max | 1.856 | 1.840 | 1.879 | 1.905 | 1.882 | 1.297 | 1.298 | 1.299 | 1.298 | 1.296 |
| Mean | 1.387 | 1.389 | 1.384 | 1.383 | 1.369 | 0.294 | 0.285 | 0.303 | 0.302 | 0.314 |
| σ | 0.290 | 0.290 | 0.285 | 0.283 | 0.281 | 0.265 | 0.253 | 0.260 | 0.267 | 0.271 |

*Statistics are calculated based on mean daily values averaged for the study period 1991-2012 at cell level.

**Table S3.** Comparisons of the spatial autocorrelation (measured by Moran's I) of modeled values among the five routing algorithms

| Algorithms | Soil saturation deficit | | | Leaf area index | | |
|---|---|---|---|---|---|---|
| | 1992 | 2005 | 1991-2012 | 1992 | 2005 | 1991-2012 |
| D8 | 0.418 | 0.414 | 0.425 | 0.417 | 0.436 | 0.424 |
| D∞ | 0.433 | 0.425 | 0.436 | 0.414 | 0.431 | 0.419 |
| RMD∞ | 0.487 | 0.479 | 0.494 | 0.445 | 0.462 | 0.451 |
| MD8 | 0.507 | 0.500 | 0.515 | 0.456 | 0.474 | 0.463 |
| MFD-md | 0.528 | 0.516 | 0.535 | 0.462 | 0.483 | 0.467 |
| p | < 0.01 | | | | | |

**Table S4.** Comparisons of cell-level $D_R$ averaged for the watershed between compared algorithms

| Algorithms compared | | Soil saturation deficit | | | Leaf area index | | |
|---|---|---|---|---|---|---|---|
| | | 1992 | 2005 | 1991-2012 | 1992 | 2005 | 1991-2012 |
| D8 | D∞ | 0.063 | 0.068 | 0.065 | 0.181 | 0.184 | 0.178 |
| D8 | RMD∞ | 0.062 | 0.067 | 0.064 | 0.174 | 0.174 | 0.177 |
| D8 | MD8 | 0.062 | 0.067 | 0.063 | 0.175 | 0.175 | 0.175 |
| D8 | MFD-md | 0.065 | 0.070 | 0.067 | 0.169 | 0.172 | 0.176 |
| D∞ | RMD∞ | 0.027 | 0.029 | 0.028 | 0.114 | 0.116 | 0.108 |
| D∞ | MD8 | 0.034 | 0.037 | 0.035 | 0.132 | 0.134 | 0.124 |
| D∞ | MFD-md | 0.043 | 0.046 | 0.044 | 0.154 | 0.156 | 0.147 |
| RMD∞ | MD8 | 0.014 | 0.015 | 0.014 | 0.061 | 0.062 | 0.054 |
| RMD∞ | MFD-md | 0.026 | 0.028 | 0.026 | 0.098 | 0.098 | 0.089 |
| MD8 | MFD-md | 0.020 | 0.022 | 0.021 | 0.080 | 0.081 | 0.072 |